# Dancing Between Success and Failure:
# Edit-level Simplification Evaluation using SALSA 🎉

**David Heineman, Yao Dou, Mounica Maddela, Wei Xu**

School of Interactive Computing

Georgia Institute of Technology

{david.heineman, douy, mmaddela}@gatech.edu; wei.xu@cc.gatech.edu

## Abstract

Large language models (e.g., GPT-4) are uniquely capable of producing highly rated text simplification, yet current human evaluation methods fail to provide a clear understanding of systems' specific strengths and weaknesses. To address this limitation, we introduce SALSA, an edit-based human annotation framework that enables holistic and fine-grained text simplification evaluation. We develop twenty one linguistically grounded edit types, covering the full spectrum of success and failure across dimensions of conceptual, syntactic and lexical simplicity. Using SALSA, we collect 19K edit annotations on 840 simplifications, revealing discrepancies in the *distribution* of simplification strategies performed by fine-tuned models, prompted LLMs and humans, and find GPT-3.5 performs more quality edits than humans, but still exhibits frequent errors. Using our fine-grained annotations, we develop LENS-SALSA, a reference-free automatic simplification metric, trained to predict sentence- and word-level quality simultaneously. Additionally, we introduce word-level quality estimation for simplification and report promising baseline results. Our data, new metric, and annotation toolkit are available at https://salsa-eval.com.

## 1   Introduction

Text simplification aims to improve a text's readability or content accessibility while preserving its fundamental meaning (Stajner, 2021; Chandrasekar et al., 1996). Traditional human evaluation for text simplification often relies on individual, shallow sentence-level ratings (Sulem et al., 2018c; Alva-Manchego et al., 2021), easily affected by the annotator's preference or bias. Maddela et al. (2023) recently proposes a more reliable and consistent human evaluation method by ranking and rating multiple simplifications altogether. However, as text simplification involves performing a series of transformations, or *edits*, such as paraphrasing, removing irrelevant details, or splitting a long sen-

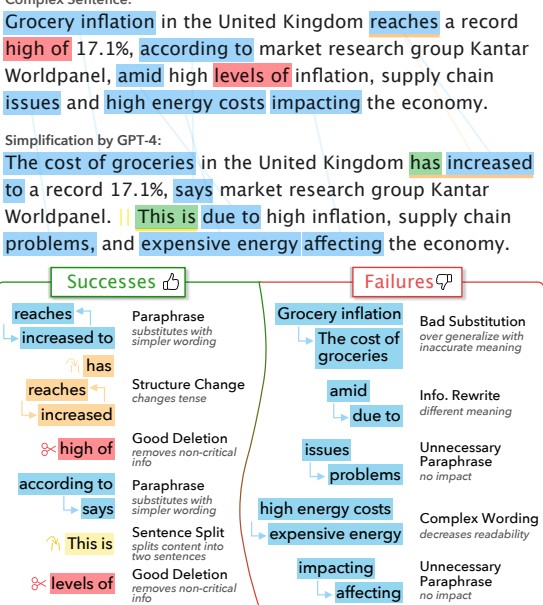

Figure 1: Simplification generated by GPT-4. Our edit-level SALSA reveals LLMs succeed across many edit types, but often fail to paraphrase and generalize.

tence into multiple shorter ones (Xu et al., 2012), sentence-level scoring remains difficult to interpret since it is not reflective of detailed information about the types of edits being performed.

Fine-grained human evaluation through span selection has been explored for machine translation (Lommel et al., 2014) and open-ended text generation (Dou et al., 2022). Yet, these evaluation methods are error-driven – i.e., focusing solely on evaluating *failure* – which punishes creative and diverse generations with minor errors in favor of generic ones. Additionally, machine translation and open-ended generation tasks usually retain none of the input words, while text simplification must balance the editing and preservation of words in the original input (Xu et al., 2016). We thus evaluate simplification quality as the aggregation of edit *successes* and *failures*, as depicted in Figure 1.

We introduce SALSA 🎉 – **S**uccess and FA**ilure**-driven **L**inguistic **S**implification **A**nnotation – an

*edit-level* human evaluation framework capturing a broad range of simplification transformations. SALSA is built on a comprehensive typology (§2) containing 21 quality and error edit types. Using SALSA, we develop an interactive interface and collect 19K edit annotations of 840 simplifications written by eleven state-of-the-art language models and two humans. With these annotations, we conduct a large-scale analysis of model and automatic metric performance, and further introduce the automatic word-level quality estimation task for text simplification. Our main findings are as follows:

- Few-shot GPT-3.5 far surpasses existing models, particularly in making syntax and content edits. However, its simplifications are not *aligned* to the types of operations performed by human. (§4)

- Some fine-tuned models such as the MUSS (Martin et al., 2022) produce more diverse edits than GPT-3.5, yet suffer from incredibly high errors, while others (T5, Raffel et al., 2020) learn to minimize loss by making very few changes. (§4)

- Open-source instruction fine-tuned models such as Alpaca (Taori et al., 2023) and Vicuna (Chiang et al., 2023) perform a similar number of edits as GPT-3.5, but at a cost of more conceptual errors due to the inherent limits of model imitation. (§4)

- Fine-tuned on SALSA annotations, our reference-free metric, LENS-SALSA, captures the subtleties of specific simplification approaches beyond existing automatic evaluation metrics. (§5)

- Leveraging our data, we present the automatic word-level quality estimation task for text simplification and establish several baseline approaches for future modeling efforts. (§6)

Our results demonstrate that SALSA provides an interpretable and exhaustive evaluation of text simplification.

## 2 SALSA 💃 Framework

We introduce SALSA, an edit-based human evaluation framework for text simplification. SALSA is defined by a typology of 21 linguistically-grounded edit types with the aim of capturing both successes and failures (i.e., quality changes and errors, see Figure 1). The annotation methodology of SALSA is structured as a decision tree and implemented via an easy-to-use interface, illustrated in Figure 2. Our interface is designed with Thresh (Heineman et al., 2023), and we release our configuration to encourage adaptation to other text rewriting tasks

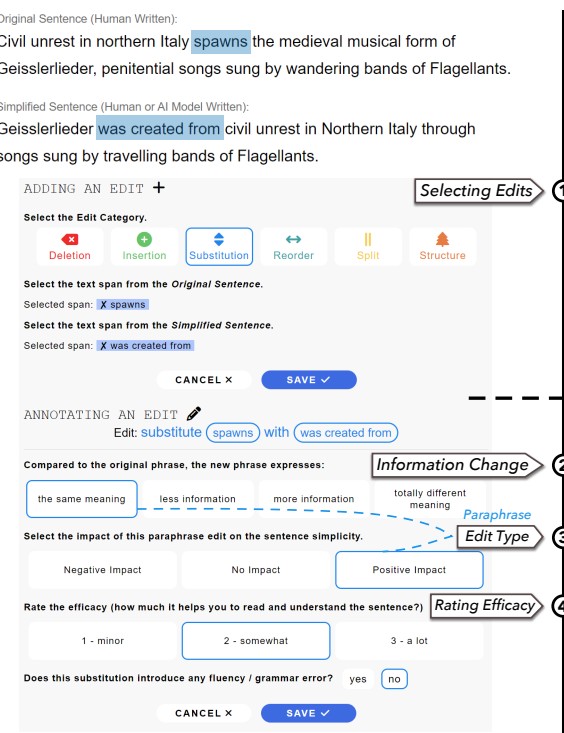

Figure 2: The SALSA annotation process consists of (1) selecting edits, (2) identifying information change, (3) classifying edit type and (4) rating efficacy/severity.

(Du et al., 2022) or collecting fine-grained human feedback (Wu et al., 2023)[1]. In the following, we describe each step of the annotation process.

### 2.1 Edit Selection

Annotation begins with *edit selection*, where annotators identify the edits performed by the simplification and select the corresponding spans for each edit. We define six types of edit operations: single-operation insertion, deletion, substitution, word-/clause-reorder, and multi-operation sentence split and structure changes. An insertion or deletion edit exclusively modifies content, while a substitution either modifies or paraphrases content. Reorder, split, or structure edits perform a context-free syntax transformation. As split and structure edits are multi-operation (i.e., require a combination of single operations), they are defined by a set of underlying single-operation *constituent* edits. For example, this structure change from passive to active voice made by zero-shot GPT-3.5 involves multiple constituent edits:

**EXAMPLE** *Zero-shot GPT-3.5*
On 14 November, an interview with journalist Piers Morgan was published, where Ronaldo said ...
On 14 November, Piers Morgan interviewed Ronaldo, who expressed ...

---

[1] https://thresh.tools/salsa

## 2.2 Categorizing by Information Change

Each selected edit is then labeled with its impact on the underlying sentence information: *less*, *same*, *more* or *different* information. Given the type of operation and change to information, we subsequently organize each edit into three linguistic families as defined by Siddharthan (2014):

**Lexical edits** perform simple changes in "wording". This includes paraphrasing (i.e., substitution that keeps the same information) and inconsequential trivial changes (e.g., inserting 'the').

**Syntax edits** capture transformations to the *distribution* of information, rather than substance. A split converts a candidate sentence to two sentences, a re-order edit re-arranges clauses or wording within a clause, and a structural edit modifies the voice, tense or clausal structure. Examples of structural edit sub-types are in Appendix B.

**Conceptual edits** modify underlying ideas conveyed by the text. A conceptual edit requires elaboration to add clarifying information or generalization to delete unnecessary/complicated ideas.

## 2.3 Edit Type Classification

After being categorized into lexical, syntax, or conceptual edit families, we further classify each edit operation into 21 fine-grained success (quality), failure (error), or trivial edit types as listed in Figure 3. *Successful edits* simplify through diverse approaches, from paraphrasing complex spans, generalization of unnecessary information, or elaboration to add clarity and background context. E.g.,

> **EXAMPLE** (*elaboration*)        *Vicuna 7B*
> ... can be fitted to an exponentially decaying curve.
> ... can be represented by a curve that gets smaller and smaller over time.

Often small edits, particularly to syntactic structure, can improve clarity, such as this addition of a clear subject-verb structure through the inclusion of the relative pronoun 'who':

> **EXAMPLE** (*structure change*)       *GPT-4*
> Paltrow in turn claims he was the one crashing rather than the other way around.
> Paltrow says he was the one who crashed, not her.

Or this conversion of the participial phrase to a relative clause to help explain significance:

> **EXAMPLE** (*structure change*)      *ChatGPT*
> ... for the first time since 2006, ending their 17-year playoff drought ...
> ... for the first time in 2006, which means they have ended their 17-year playoff drought.

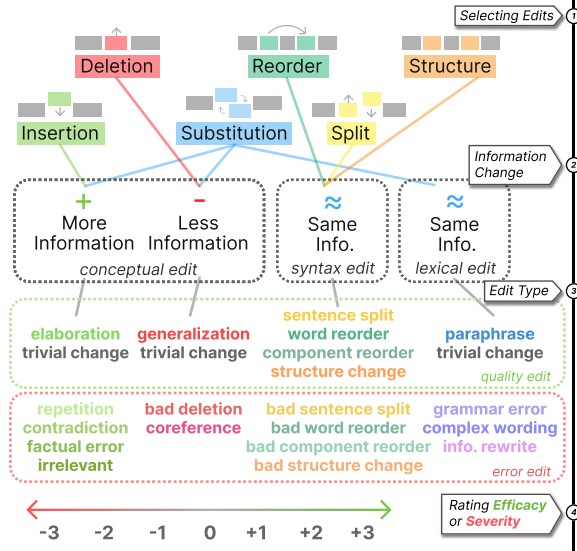

Figure 3: The multi-stage SALSA edit evaluation framework. Spans are classified into twenty one success and failure types (*trivial change* counts as one type) using the interface shown in Figure 2.

Sentence splitting or reordering information may clarify a sequence of events:

> **EXAMPLE** (*component reorder*)     *ChatGPT*
> Poland announces the closure of a major border crossing with Belarus "until further notice" amid heightened tensions between the two countries.
> Poland has closed a big border crossing with Belarus due to increased tensions between the two countries. The closure will remain in effect until further notice.

*Failure edits* include any ablation from minor readability issues to hallucinations or deletions to sentence meaning. In the following example, the *coreference error* captures the deleted reference between the 'ICJ' and 'US' acronyms to their original definitions, useful contextual information:

> **EXAMPLE** (*coreference error*)     *ChatGPT*
> The International Court of Justice (ICJ) rules that the United States violated its ...
> The ICJ said that the US broke its ...

And often multiple edits overlap, such as this *information rewrite* which successfully adds clarity via reordering, but botches the author's sarcasm:

> **EXAMPLE** (*information rewrite*)    *Alpaca 7B*
> ... justifies a runtime nearing 3 hours (with a post-credits scene, no less), and it already opened to over $100 million worldwide.
> .. takes up almost 3 hours of the movie. The movie opened to over $100 million worldwide. A post-credits scene completes the story.

We also separately ask annotators to identify if the edit contains a *grammar error*. Appendix A provides an exhaustive description and examples for each edit type.

## 2.4 Rating Edit Efficacy / Severity

As each edit has a varying degree of impact on overall simplification quality, we finally ask annotators to rate the efficacy of quality edits or severity of error edits. We define three levels: 1 – minor, 2 – somewhat, and 3 – major. Examples of each severity level are included in Appendix A.3.

## 3 Data Collection

We describe our use of SALSA to collect 19K edit annotations covering 11.6K spans on 840 model-generated and human-written simplifications.

### 3.1 Simplification Data

Data collection is performed on an extended version of SIMPEVAL₂₀₂₂ (Maddela et al., 2023), including a train set covering state-of-the-art simplification systems and held-out test set of recent LLMs. We include a full description of each system in Appendix C.1.

**SALSA Train.** We first extend the 360 simplifications from SIMPEVAL₂₀₂₂ to 700 simplifications based on 100 complex sentences from Wikipedia articles dated between Oct 2022 and Dec 2022. The complex sentences are unseen during the training of the LLMs and were selected to be intentionally difficult (avg. length of 37.3 words) to enable an evaluation of the models' full capabilities in performing diverse simplification edits. Simplifications are generated by five models including fine-tuned T5-3B and T5-11B (Raffel et al., 2020), MUSS (Martin et al., 2022), a controllable BART-large model trained with unsupervised, mined paraphrases, zero- and few-shot GPT-3.5 (Ouyang et al., 2022), and two human-written references. For modeling experiments in §5 and §6, we divide the initial 700 simplifications by the complex sentence with a 70/30% train/dev split.

**SALSA Test.** We further gather 20 more complex sentences from Wikipedia articles published in Mar 2023 and generate 140 simplifications using recent LLMs including GPT-3.5, ChatGPT, GPT-4, Alpaca-7B (Touvron et al., 2023) and Vicuna-7B (Chiang et al., 2023), along with T5-3B and T5-11B fine-tuned with control tokens.

### 3.2 Annotation

As crowd-sourced annotators have shown to have inconsistent quality (Shmueli et al., 2021), we hire 6 undergraduate students from a US university. Annotators were trained with an in-depth tutorial con-

| Edit | Sub-type | Kripp. $\alpha$ | 3 Agree% | 2 Agree% |
|------|----------|---------|----------|----------|
| Insertion | More Information | 0.45 | 14% | 40% |
| Deletion | Less Information | 0.75 | 42% | 65% |
| Substitution | More Information | 0.15 | 1% | 11% |
| | Less Information | 0.31 | 7% | 26% |
| Reorder | Word-level | 0.12 | 0% | 13% |
| | Component-level | 0.41 | 11% | 38% |
| Split | Sentence Split | 0.66 | 32% | 55% |
| Structure | Structure | 0.25 | 5% | 25% |
| Substitution | Same Information | 0.53 | 21% | 51% |

Table 1: Edit selection inter-annotator agreement measured per token. As Krippendorff's $\alpha$ (2018) includes unlabeled tokens, we also report the percentage of annotated tokens where at least 2 and 3 annotators agree.

sisting of broad explanations of simplification concepts, over 100 examples covering each of the 21 SALSA edit types and interactive exercises, completed two rounds of onboarding annotations and were provided continuous feedback by the authors. To concretely measure agreement for each stage of the SALSA framework, we collect annotations in three stages: (1) we have three annotators select edits, (2) a fourth annotator adjudicates the edits into a single selection and (3) the initial three annotators classify and rate the adjudicated edits. Figure 2 illustrates our annotation interface, with further screenshots of our tutorial included in Appendix G.

### 3.3 Inter-Annotator Agreement

We calculate edit selection agreement (i.e. agreement prior to adjudication) by each token, with Table 1 reporting agreement per edit, further broken down by their type of information change. We observe edit agreement is highly dependent on the edit type and type of information change being performed. High agreements are seen for deletion ($\alpha$=0.75), paraphrase (substitution with the same information, $\alpha$=0.53), and sentence splits ($\alpha$=0.66). Substitution that introduces more information, however, exhibits lower agreement ($\alpha$=0.15), due to the subjectivity among annotators on determining whether new tokens contain 'novel' information, as was often mixed up with insertion. Reordering ($\alpha$=0.12) and structure edits ($\alpha$=0.25) also report lower agreements. We fully explore the phenomenon of annotator disagreement in Appendix C.2, and find overlapping syntactic and content edits often have multiple correct interpretations, leading to an inherent disagreement. Additionally, we find our % rates for annotator agreement are similar to fine-grained evaluation frameworks in other text generation tasks (Dou et al., 2022).

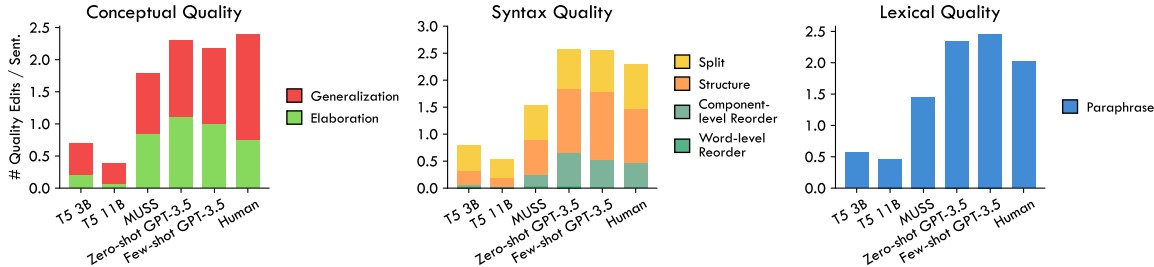

Figure 4: Successful edits per-model, organized by edit type. MUSS outperforms fine-tuned T5 but fails to capture more complex simplification techniques. Compared to GPT-3.5, human written simplifications have more generalization ■, a similar distribution of syntax edits, and slightly less paraphrasing ■.

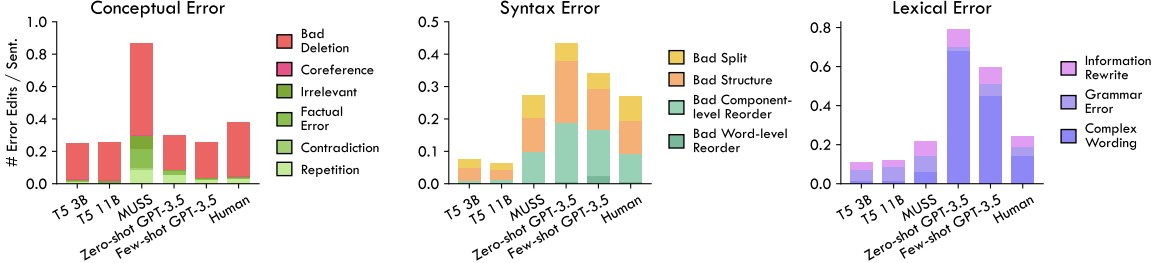

Figure 5: Failure edits per-model, organized by edit type. Compared to humans, both GPT-3.5 setups make more syntax and lexical errors. Although humans perform bad deletion ■ errors at a higher frequency than GPT-3.5, this is reflective of the inherent ambiguity in judging the relevancy of the deleted content.

## 4 Key Analysis

We use SALSA to evaluate state-of-the-art simplification by collecting annotations on our extended version of the SIMPEVAL corpus (Maddela et al., 2023), which includes fine-tuned, LLM- and human-written simplifications. Our resulting data collection includes 19K edit annotations across 840 simplifications.

We present our primary results in Figures 4, 5, and 6. Figures 4 and 5 illustrate the frequency of quality and error edit types. As edits vary in length, we calculate *edit coverage*: the length of each edit in proportion to the total length of the simplification and report the average edit coverage for different efficacy and severity ratings in 6, showing a view of edit ratings adjusted for length. Additionally, we include Figure 7, which compares simplifications generated by recent instruction fine-tuned language models. The following are our key findings:

**Models primarily write good edits, but still trail humans (Fig. 4, 5).** We observe that 16% of model-generated edits are errors, with the best-performing model, few-shot GPT-3.5, producing errors in only 9% of edits. We find this still trails human simplifications, which have an error rate of 6%. MUSS and GPT-3.5 have a median count of 1 error per simplification and 63% of their simplifications contain at least one error, showing these errors are not concentrated in a few 'bad' simplifications but instead

often occur among many good edits.

**Language models elaborate, while humans generalize (Fig. 4).** When simplifying content, all models (excluding T5) tend to elaborate at a higher ratio than humans, for example, GPT-3.5 attempts to insert content 17% more often. As LLMs have shown to encode world knowledge in their parameters (Petroni et al., 2019; Brown et al., 2020), GPT-3.5 elaboration is far more effective than MUSS, for example:

> **EXAMPLE** *Few-shot GPT-3.5*
> After defeating PSD candidate Viorica Dăncilă by a landslide in 2019, **his** second term..
>
> In 2019, Klaus Iohannis defeated PSD candidate Viorica Dăncilă by a large margin. His second term..

**GPT-3.5 writes quality edits at a higher frequency than humans, but human edits are longer and more effective (Fig. 4, 6).** Both zero-shot and few-shot GPT-3.5 produce a larger number of edits, but human edits are more substantial, as demonstrated by the higher edit coverage across all efficacy levels, particularly for syntax and lexical edits. Human simplification typically deletes, paraphrases, or reorders entire clauses, while GPT-3.5 often edits single modifiers or words.

**Fine-tuned T5-3B and T5-11B generate conservative simplifications (Fig. 4, 5, 6).** Compared to all other systems, both T5 models make minimal changes in terms of frequency and edit coverage,

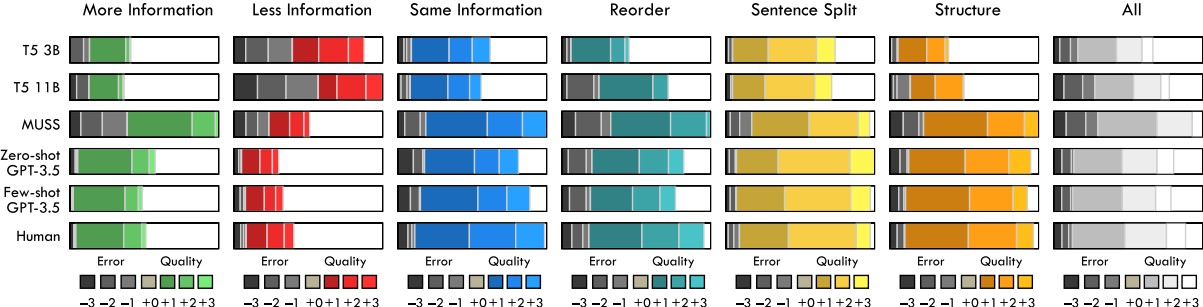

Figure 6: Edit coverage of efficacy (+) and severity (-) ratings for each model, separated by simplification approach, with edit coverage defined as $(len(e_C) + len(e_S))/(len(C) + len(S))$ (see §A.4). Overall, humans make the longest quality edits and most infrequent error edits. We report the distribution of each edit rating in Figure 14.

while still exhibiting high rates of error. This is likely due to their training data, Wiki-Auto (Jiang et al., 2020), containing shorter sentences, usually requiring simpler simplification techniques, making it difficult for models to generalize on longer and more complex sentences. Later in Appendix D, we show using control tokens (Martin et al., 2020) during training, as done by MUSS, can improve diversity but at the expense of increasing deletion and hallucination errors.

**Split edits are straightforward, Structure edits are far more complex (Fig. 4, 5).** Surprisingly, sentence splitting is shown to be the easiest edit for all models to accomplish, with a similar number made by MUSS, GPT-3.5, and humans, with even the conservative T5 models making a comparable number of split edits. However, structure change and re-ordering edits are rarely seen in fine-tuned models. We speculate this may be attributed to (i) these types of edits are infrequent in the training data and (ii) GPT-3.5 has a unique ability to perform complicated syntax rewriting, echo with the findings in abstractive summarization (Goyal et al., 2022). Despite GPT-3.5's improvement, the structure error rate demonstrates it has not yet reached human-level ability. Additionally, we observe zero-shot GPT-3.5 produces structure errors (see below example) at a 19% rate higher than few-shot.

> **EXAMPLE** *Zero-shot GPT-3.5*
> The sentence included a fine of $400...
> You will receive a fine of $400...

We find human simplifications are more conservative with re-ordering than models, yet attempts to simplify with re-ordering often appear arbitrary:

> **EXAMPLE** *Human written*
> On 3 November 2022, the British Secretary...
> On November 3rd, 2022, the British Secretary...

**Humans appear to produce bad deletion errors, but these are often subjective (Fig. 5).** *Bad dele-*

*tion* constitutes 35% of error edits made by humans, compared to 8% by few-shot GPT-3.5. The anomaly of the *bad deletion* errors reveals an inherent subjectivity in assessing deletion:

> **EXAMPLE** *Human written*
> Unlike the first film adaptation, in which director Samuel Fuller removed...
> Unlike the first film adaptation, Samuel Fuller removed...

In this example, some annotators marked the edit as a bad deletion while others consider it appropriate. As the sentence discusses a book adaptation into a film, the description of 'Samuel Fuller' is helpful depending on the reader, which underscores the need for adaptive levels of simplification to accommodate each reader's needs.

**Paraphrasing is a crucial, but tricky mechanism (Fig. 4, 5).** MUSS, GPT-3.5, and humans all paraphrase in at least 75% of sentences. Despite low performance in conceptual and syntactic simplification, MUSS paraphrases at a human-like rate likely due to its training on over one million paraphrase sentence pairs mined from web crawl data. Although zero-/few-shot GPT-3.5 paraphrases at a higher rate than humans, these edits are often are unnecessary. For instance:

> **EXAMPLE** *Few-shot GPT-3.5*
> The club said on social media that customers subdued the gunman...
> The club reported on social media that customers were able...

**Open-source LLMs are approaching GPT-3.5 simplifications, or are they (Fig. 7)?** Given recent attention to ChatGPT (OpenAI, 2022), GPT-4 (OpenAI, 2023), and the emergence of instruction fine-tuning smaller language models on outputs from proprietary LLMs, we perform a supplementary evaluation on these systems. The open-source Alpaca (Taori et al., 2023) and Vicuna (Chiang et al., 2023) appear to perform a similar number of

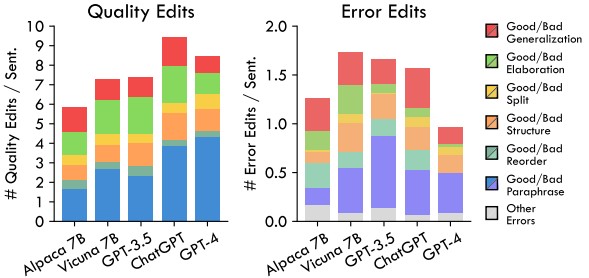

Figure 7: Success and failure edits on simplifications by five recent instruction fine-tuned language models.

quality and error edits to GPT-3.5. However, these systems tend to write far more bad elaboration errors such as factual errors or contradictions:

> **EXAMPLE** *Alpaca 7B*
> ... a controversial "angel tax" provision seeking to capture some of the income entering the country from foreign investors funding India's start-ups.
>
> ... a controversial "angel tax" provision, which is aimed at stopping foreign investors from funneling money into India's startups.

This behavior suggests open-source instruction fine-tuned models mimic the style of their larger counterparts, but not their knowledge, a phenomenon observed by Gudibande et al. (2023). GPT-4 exhibits the best performance by making fewer content errors while producing a high number of quality edits, but still exhibits errors particularly when paraphrasing individual spans without considering the broader sentence meaning:

> **EXAMPLE** *GPT-4*
> Grocery inflation in the United Kingdom reaches a record high of 17.1% ...
>
> The cost of groceries in the United Kingdom has increased to a record 17.1% ...

While GPT-4 successfully paraphrases inflation by relating to cost, it fails to recognize the sentence is discussing inflation *rate*, rather than exact prices.

We include further analysis, discussion, and dataset statistics in Appendix D.

## 5 Evaluating Metric Edit Sensitivity

While automatic metrics are traditionally evaluated using correlation with sentence-level, Likert scale human ratings on dimensions of adequacy, fluency and simplicity, this fails to understand the ability of automatic metrics to capture the subtleties of lexical, syntactic, and conceptual simplification. With our SALSA annotations, we study how well current automatic metrics capture these distinct simplification approaches. Additionally, we introduce LENS-SALSA, a reference-free metric fine-tuned on SALSA annotations.

| | | BLEU | SARI | BERTSCORE | COMET-MQM | LENS | LENS-SALSA |
|---|---|---|---|---|---|---|---|
| Quality | Lexical | -0.167 | 0.126 | 0.025 | 0.120 | 0.407 | **0.443** |
| | Syntax | 0.013 | 0.204 | 0.147 | 0.122 | 0.306 | **0.356** |
| | Conceptual | 0.043 | 0.149 | 0.097 | 0.038 | 0.144 | **0.202** |
| Error | Lexical | -0.147 | -0.026 | -0.093 | -0.068 | -0.041 | **0.054** |
| | Syntax | -0.104 | -0.013 | -0.043 | -0.017 | 0.019 | **0.086** |
| | Conceptual | 0.047 | 0.150 | **0.279** | 0.228 | 0.207 | 0.107 |
| All | All Error | -0.121 | 0.067 | 0.117 | 0.127 | 0.161 | **0.169** |
| | All Quality | -0.095 | 0.179 | 0.027 | 0.074 | 0.336 | **0.459** |
| | All Edits | -0.116 | 0.170 | 0.056 | 0.092 | 0.334 | **0.446** |

Table 2: Pearson correlation between automatic metrics and SALSA sub-scores (§A.4) on the SALSA test set. All reference-based metrics use two human-written references. **Best**; Second Best.

**Existing Automatic Metrics.** We consider five automatic metrics: BLEU (Papineni et al., 2002), SARI (Xu et al., 2016), the most widely-used text simplification metric, BERTSCORE (Zhang et al., 2020), COMET-MQM, a machine translation metric (Rei et al., 2020) trained on MQM ratings (Freitag et al., 2021), and LENS (Maddela et al., 2023), a recently proposed text simplification metric fine-tuned on SIMPEVAL that contains rank-based human ratings of simplifications from 24 systems.

**LENS-SALSA.** The automatic simplification metrics mentioned above require human-written references, which may not be available in every evaluation setting. To this end, we introduce LENS-SALSA, a reference-free simplification metric enabled by edit-level information. Based on the COMETKIWI machine translation metric design (Rei et al., 2022), we first pre-train LENS-SALSA on the sentence-level human ratings from SIMPEVAL using UniTE (Wan et al., 2022), a multi-task learning method. Specifically, the metric is trained on the same score but from three input formats: *Simp:Ref*, *Simp:Complex*, and *Simp:Complex:Ref*, where ":" denotes concatenation. Then, we fine-tune LENS-SALSA on SALSA annotations using a dual-objective to predict both the sentence-level score (calculated by LENS) and a word-level quality score $\hat{w}_i \in [-3, 3]$, corresponding to the efficacy or severity rating (§2.4) of each word $w_i$ in the complex and simplified sentences. We use RoBERTa-large as the base model for LENS-SALSA, and 490, 210, and 140 sentence pairs for train, validation, and test, respectively. Implementation details are provided in Appendix F.2.

**Results.** As fine-grained MQM annotations in machine translation are considered a gold-standard in metric evaluation (Freitag et al., 2021), we adapt their method (detailed in §A.4) to collapse edit-

level ratings to a single score, and calculate sub-scores by only considering certain edit types. Table 2 reports the Pearson correlation between metric scores and human sub-scores across each SALSA dimension. LENS-SALSA achieves the highest correlation in nearly all edit approaches, showing its capability to capture all forms of simplification. Overall, only LENS and LENS-SALSA obtain substantial correlation with the overall human SALSA scores (0.33 and 0.45 respectively), while other metrics have spurious and even negative correlations with human judgments. Interestingly, COMET-MQM, intended for machine translation, performs better than BLEU and BERTScore, which further underlines the value of span-based ratings for trained metrics. Despite strong performance, we find LENS mainly evaluates lexical and syntactic edits, rather than conceptual ones, which may be attributed to its training data consisting of shorter, paraphrase-based simplifications. Lastly, all metrics have substantially higher correlation with quality than error edits. We posit this is primarily due to the sparsity and wide range of errors exhibited in the generations of current high-performing systems.

## 6 Word-Level Quality Estimation

Word-level quality estimation (QE) is the task of predicting the quality of each token in a generation, and has substantial downstream application to evaluating and refining text simplification. Despite word-level QE being a well understood task in machine translation (Basu et al., 2018; Zerva et al., 2022), it has not yet been studied for text simplification due to a lack of appropriately annotated data. In this section, we use SALSA annotations to demonstrate baseline approaches and highlight potential for future work.

**Task.** We define word-level simplification QE as classifying each token in the complex and simplified sentences as *quality*, *error*, or *ok*. To adapt SALSA for the QE task, we label each token by the average efficacy/severity rating of its associated edit: $< 0$ as error, $= 0$ as ok, and $> 0$ as quality. Words that are not part of any edits default to the ok label. We deconstruct split and structure edits into their constituent edits, only label the simplified spans for substitution edits, and exclude reorder edits due to their low frequency. The final label counts for our train, validation, test splits are: 6.8K/1.8K/27K, 2.7K/627/11K, and 1.7K/484/6.9K for quality/error/ok respectively.

| Method | Quality | Error | Ok | Average |
|---|---|---|---|---|
| *End-to-end* | | | | |
| Tag | 67.00 | 28.24 | 92.88 | 62.71 |
| Tag-ML | **70.73** | 30.06 | **93.09** | 64.62 |
| *Two-stage (use word aligner to get edit information)* | | | | |
| Tag-EI | 69.09 | 30.37 | 93.04 | 64.17 |
| Ec-Sep | 64.87 | 36.15 | 91.56 | 64.20 |
| Ec-One | 68.77 | **39.50** | 91.91 | **66.73** |
| Oracle (Ec-One) | 88.31 | 69.44 | 98.35 | 85.47 |

Table 3: Word-level F1 scores of different methods on SALSA test set. Oracle uses annotated edit information.

**Methods.** We propose two approaches: End-to-end, where a single model labels each token directly; and Two-stage, where a word aligner first identifies edits, then the model labels each token using the identified edit information. For end-to-end, we implement the following two methods:

*Tagging (Tag)* is a native sequence tagging model with a classification head.

*Tagging with Multi-task Loss (Tag-ML)* is similar to the tagging method except trained with a multi-task loss function: $\mathcal{L} = \mathcal{L}_{tag} + \mathcal{L}_{ec}$. $\mathcal{L}_{ec}$ is an additional objective that classifies each token into *none*, *deletion*, *substitution*, or *insertion*.

For two-stage methods, we first apply a QA-based word aligner (Nagata et al., 2020) to the sentence pair and use a set of rules to convert word alignments to edits: consecutive non-aligned words in the original sentence are labeled as a deletion edit; consecutive non-aligned words in the simplified sentence are labeled as an insertion edit; and aligned words or spans that differ are labeled as a substitution edit. Here are three two-stage methods:

*Tagging with Edit Information (Tag-EI)* is a sequence tagging model with a classification head that takes the concatenation of the hidden states of both edit type and token as the input. The hidden states of the edit type are obtained via a linear layer.

*Edit Classification with Separate Classifiers (Ec-Sep)* contains one classifier for each of the three edit operations. Each classifier is an encoder model with a feedforward neural network (FNN). The inputs to these FNNs are the hidden states of the [CLS] token and the max-pooled tokens from the edit spans (i.e., for substitution edit, one from the original span, and one from the simplified span).

*Edit Classification with One Classifier (Ec-One)* is one classifier with three FNNs mentioned above. The difference is the encoder is trained collectively.

All methods (including the word aligner) use RoBERTa-large. Further implementation details and results are included in Appendix F.

**Results.** Table 3 shows the test set performance for each label. Among the end-to-end methods, training with multi-task loss results in improvement on all three label F1 scores, achieving the second-best average F1 score overall. We find edit classification approaches detect error tokens more accurately than tagging approaches. Within edit classification methods, using one classifier outperforms multiple ones due to the benefit of joint encoder training. Overall, the edit classification with one classifier method performs the best with a gain of over 11 points on error F1 and a 4-point increase in average F1, compared to the base tagging model.

## 7 Related Work

**Model Evaluation.** Simplification work broadly agrees some typology of simplification operations exists (Siddharthan, 2014), starting with early rule-based systems which explicitly defined specific syntax operations (Dras, 1999). Past work has experimented with designing models to control the extent of each operation by using a pipeline to perform simplification operations independently (Maddela et al., 2021; Raffel et al., 2020), predicting edit operations (Dong et al., 2019) or augmenting fine-tuned models with learned control tokens (Martin et al., 2020, 2022). However, evaluation only considers a sentence in its entirety rather than rating individual operations, either by automatic metrics (Kriz et al., 2020), shown to be an inadequate representation of quality (Alva-Manchego et al., 2021; Sulem et al., 2018a), or by surface-level Likert ratings, typically asking crowd-sourced annotators to rate on scales of fluency, adequacy, and simplicity. These scores are difficult to interpret and capture no detail into the type of simplification being written (Briakou et al., 2021; Hashimoto et al., 2019). Additionally, despite current systems' often producing simplification errors (Choshen and Abend, 2018), annotating error has primarily been performed through inspection, and has not been incorporated into human or automatic evaluation (Gooding, 2022).

**Linguistic Inspection.** Manual inspection attempts to understand the behavior of simplification models or datasets, characterized by detailed typologies and often conducted by authors or domain experts. Cardon et al. (2022) performs detailed inspection of the ASSET simplification test corpus (Alva-Manchego et al., 2020a) to study the behavior of automatic metrics and Cumbicus-Pineda

et al. (2021a) propose a framework for evaluating success and failure by answering a series of checklist items, with sentences given a capability score based on the number of requirements fulfilled. Yamaguchi et al. (2023) annotates simplifications of earlier models such as DRESS (Zhang and Lapata, 2017) and SUC (Sun et al., 2020) using a taxonomy of 62 error categories, but do not analyze the SOTA, MUSS, or LLMs. Stodden and Kallmeyer (2022) proposes an interactive linguistic inspection interface, but this interface is not designed for human evaluation of model outputs and does not provide ratings for measuring performance.

**Fine-grained Human Evaluation.** Human evaluation performed on a span-level has been previously proposed for a variety of NLP tasks. In translation, the Multidimensional Quality Metrics (MQM) (Lommel et al., 2014), categorizes error into accuracy and fluency sub-types and is later extended by Freitag et al. (2021) to weight errors by severity and combine into a single quality score. Dou et al. (2022) proposes SCARECROW to capture errors appearing in open-ended text generation. However, as these span-based evaluation schemes exclusively annotate error, they encourage generic outputs and punish interesting or diverse generations. For summarization, the FRANK typology (Pagnoni et al., 2021) aggregates errors into broader categories to benchmark metrics that measure factuality. Inspired by FRANK, Devaraj et al. (2022) introduces a framework to evaluate factuality for text simplification.

## 8 Conclusion

In this work, we introduce SALSA, a novel edit-based evaluation framework incorporating error and quality evaluation, and dimensions of lexical, syntax and conceptual simplification and demonstrate SALSA benefits in granularity, accuracy, and consistency. We employ SALSA to collect a 19K edit annotation dataset and analyze the strengths and limitations of fine-tuned models, prompted LLMs, and human simplifications. Finally, we use SALSA annotations to develop a reference-free automatic metric for text simplification and demonstrate strong baselines for word-level quality estimation, showing promising avenues for the development of fine-grained human evaluation.

## Limitations

Our annotation only represents a single use case of text simplification and we encourage an extension of SALSA to domain-specific simplification, such as medical (Joseph et al., 2023), legal (Garimella et al., 2022), or multi-lingual text (Ryan et al., 2023), and annotations by groups of specific downstream users (Stajner, 2021). The LENS-SALSA reference-free metric is trained exclusively on Wikipedia simplification, and we do not consider its cross-domain generalization or its ability to capture the simplification need to specific target communities. Additionally, while we demonstrate promising results on sentence-level evaluation, simplification is often a document-level task (Laban et al., 2021; Sun et al., 2021). Incorporating higher-level operations such as sentence fusion, paragraph compression, and reordering would require an extension to SALSA and presents unique analytical challenges. Finally, detailed human evaluation inherently requires greater resources to produce a high granularity of annotations. While we show this process can be streamlined with a robust annotator training, SALSA requires a similar amount of resources as widely used fine-grained evaluation in other tasks such as MQM (Lommel et al., 2014) or FRANK (Pagnoni et al., 2021).

## Ethics Statement

Our annotations were performed using the SIMPE-VAL$_{2022}$ corpus, originally collected from publicly available Wikipedia articles (Maddela et al., 2023) and we further extend the dataset with complex sentences collecting using the same methodology from publicly available Wikipedia articles. As discussed in §3.2, we perform data collection with in-house annotators from a US university. Annotators were all native English speakers and paid $15-$18/hour. We took care to manually review all data prior to annotation as to exclude any triggering or sensitive material from our annotation data. Annotators were informed that any data they felt uncomfortable with was not required to annotate. Our interface was built using the open-source Vue.js[2] library, and training of our added T5-11B system was implemented using the open-source Hugging Face Transformers[3] library.

---

[2]https://vuejs.org/
[3]https://huggingface.co/

## Acknowledgements

We thank Tarek Naous, Nghia T. Le, Fan Bai, and Yang Chen for their helpful feedback on this work. We also thank Marcus Ma, Rachel Choi, Vishnesh J. Ramanathan, Elizabeth Liu, Govind Ramesh, Ayush Panda, Anton Lavrouk, Vinayak Athavale, and Kelly Smith for their help with human annotation. This research is supported in part by the NSF awards IIS-2144493 and IIS-2112633, ODNI and IARPA via the HIATUS program (contract 2022-22072200004). The views and conclusions contained herein are those of the authors and should not be interpreted as necessarily representing the official policies, either expressed or implied, of NSF, ODNI, IARPA, or the U.S. Government. The U.S. Government is authorized to reproduce and distribute reprints for governmental purposes notwithstanding any copyright annotation therein.

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

# A  Defining the SALSA 🪇 Framework

We provide detail into the SALSA framework, including qualitative examples which helped guide design decisions when building the typology. Table 4 illustrates each final edit type, as organized by Figure 3. During development, we adjusted our scheme based on preliminary annotations with the final goal of SALSA's ability to evenly represent all modes of simplification and the full space of errors.

## A.1  Quality Evaluation

We organize quality edits by their approach to simplification, as real-world application and models' capability to simplify falls into tiers of *conceptual*, *syntactic* and *lexical* simplification (Stajner, 2021). An ideal simplification system demonstrates a balance of these 'tiers' and incorporates different techniques depending on the original text, context and users (Gooding and Tragut, 2022). Automatic simplification research initially focused on lexical paraphrasing (Siddharthan, 2014), but has since evolved to emphasize the importance of syntactic and conceptual editing (Alva-Manchego et al., 2020b).

### A.1.1  Conceptual Simplification

These edits modify the underlying sentence information or ideas, a prerequisite for simplifying complex domains. We consider 'conceptual simplification' to be interchangeable with 'semantic simplification' as used in some literature (Sulem et al., 2018b; Jiang et al., 2022).

**Elaboration.** An addition of meaningful, relevant and correct information (Siddharthan, 2006), such as clarifying vague terminology, providing background information on an entity or subject, or explicating general world knowledge unknown to the audience. Elaboration has been shown as a rare, but helpful mechanism in text generation (Cao et al., 2022) and we observe its careful use in human simplifications.

**Generalization.** A deletion of unnecessary, irrelevant or complicated concepts. Although we ask annotators to rate the quality of elaboration by how it *improves the readability* of a sentence, we ask annotators to rate the quality of a generalization by the *relevancy of the deleted information to the main idea* of the sentence. As 'relevancy' is inherently subjective to the user, domain and annotator, determining the threshold for 'necessary information' is crucial to standardize (Devaraj et al., 2022).

Deleting information will, by nature, contain some amount of information and SALSA instead focuses on ensuring the deleted information is not important sentence, context or users. Consider two candidate deletions:

> **EXAMPLE**
> Like so many hyped books before it, *The Midnight Library* excited me and gave me pause.
> Like so many hyped books before it, *The Midnight Library* excited me and gave me pause.

Although the deletion of *Midnight* is shorter, it changed the subject of the sentence, and it is rated higher than the second deletion, which is not central to the main idea. Generalization using paraphrase is more often preferred than deleting full clauses.

We observe successful conceptual edits are often performed on the clause level. For example, adjunct removal via deletion:

> **EXAMPLE**
> Born into slavery in 1856, Booker T. Washington became an influential African American leader.
> Booker T. Washington became an influential African American leader.

Or information insertion through an appositive or relative clause, although the prior is typically more common for the SIMPEVAL domain as it implies objective information:

> **EXAMPLE**
> Éric Gauthier is also a novella author...
> Éric Gauthier, famous for his soloist dancing career, is also a novella author...

### A.1.2  Syntactic Simplification

Syntax is a crucial mechanism for fluent, highly modified simplification (Štajner, 2016). Given recent attention in automatic simplification to syntax-aware datasets and systems (Cumbicus-Pineda et al., 2021b; Kumar et al., 2020; Alva-Manchego et al., 2020a; Scarton et al., 2017), SALSA standardizes the first explicit evaluation accounting for these operations.

**Information Reorder.** We classify two levels of reorder, *word-level* reorder, which reorganizes modifiers within a phrase, and *component-level* reorder which moves clauses or content across a sentence (Siddharthan, 2006). A component-level re-order typically may be accompanied by a broader structure change or both re-order types may overlap, as in:

> **EXAMPLE**
> The emergence of huge radio conglomerates is a direct consequence of the '96 Act.

| | Type | Description | Example |
|---|---|---|---|
| **Quality Evaluation** | | | |
| *Conceptual* — Elaboration | Elaboration | Meaningful and correct information which enumerates the main idea | Many volatile organic chemicals, which harm our environment, are increasing in abundance in the lower troposphere. |
| Generalization | Generalization | Removes unnecessary, irrelevant or complicated information | Many volatile organic chemicals are increasing in the lower troposphere. *(in abundance was removed)* |
| *Syntax* — Word-level Reorder | Word-level Reorder | Order of words within a phrase is swapped | Many organic volatile chemicals are increasing in abundance in the lower troposphere. |
| Component-level Reorder | Component-level Reorder | Order of phrases within a sentence is swapped | In the lower troposphere, many volatile organic chemicals are increasing in abundance. |
| Sentence Split | Sentence Split | Independent information converted to two separate sentences. | Many volatile organic chemicals are increasing. They are found in abundance in the lower troposphere. |
| Structure Change | Structure Change | Rewrites voice, tense or structure. See Appendix B for details and sub-types | The abundance of many volatile organic chemicals is increasing in the lower troposphere. |
| *Lexical* — Paraphrase | Paraphrase | Lexical complexity of the phrase decreases, while the meaning is unchanged | Many volatile organic chemicals are being seen more in the lower troposphere. |
| Trivial Change | Trivial Change | Adds clarity or removes verbosity, while the lexical complexity and meaning is unchanged | Many volatile organic chemicals are currently increasing in abundance in the lower troposphere. |
| **Error Evaluation** | | | |
| *Conceptual* — Bad Deletion | Bad Deletion | Deleted necessary and relevant content | Many chemicals are increasing in abundance in the lower troposphere. *(volatile organic was removed)* |
| Coreference | Coreference | A reference to a named entity critical to understanding the main idea is removed | They are increasing in abundance in the lower troposphere. |
| Repetition | Repetition | Phrase added or changed but fail to contain novel information or insight | Many volatile organic chemicals, which are chemicals, are increasing in abundance in the lower troposphere. |
| Contradiction | Contradiction | Phrase added or changed but clearly contradicts information presented in the original sentence | Many volatile organic chemicals, which are decreasing in our troposphere, are increasing in abundance in the lower troposphere. |
| Factual Error | Factual Error | Externally verifiable incorrect claim is made by the phrase | Many volatile organic chemicals are increasing in abundance in the lower troposphere when they decide to. |
| Irrelevant | Irrelevant | New information is introduced which is unrelated to the main idea | Many volatile organic chemicals, unlike low vapor pressure chemicals, are increasing in abundance in the lower troposphere. |
| *Syntax* — Bad Word-level Reorder | Bad Word-level Reorder | Presented a new word order with less clarity within a clause | Many volatile organic chemicals are having their abundance increasing in the lower troposphere. |
| Bad Component Reorder | Bad Component Reorder | Presented a new clausal order with less clarity | In abundance in the lower troposphere, many volatile organic chemicals are increasing. |
| Bad Structure | Bad Structure | A failed attempt to modify the voice, tense or structure | Many volatile organic chemicals have been increasing in abundance in the lower troposphere. |
| Bad Split | Bad Split | Split at an inappropriate location or interrupted the flow of ideas | Many volatile organic chemicals are increasing. They are increasing in abundance in the lower troposphere. |
| *Lexical* — Complex Wording | Complex Wording | Lexical complexity of the phrase increases, while the meaning is retained | Many volatile organic chemicals are proliferating throughout the lower troposphere. |
| Information Rewrite | Information Rewrite | All information was removed from the phrase and replaced with new information | Many volatile organic chemicals are decreasing in abundance in the lower troposphere. |
| Grammar | Grammar | Violation of conventional grammar | Many volatile organic chemicals which are increasing in abundance in the lower troposphere. |

Table 4: Overview of the SALSA edit-level evaluation typology. Original text for the examples: *Many volatile organic chemicals are increasing in abundance in the lower troposphere.*

The '96 Act had a direct consequence of the emergence of huge radio conglomerates.

When faced with two equivalent phrases (e.g. 'A and B' → 'B and A'), SALSA classifies the reordered span as the phrase more significant to the main idea of the sentence. In practice, we found this to be a helpful guideline, although annotators often simply selected the phrase appearing first in the candidate sentence.

**Structural Change.** As this syntax modification necessarily includes some *discourse preserving edits* (Gooding, 2022), they are defined w.r.t. some combination of *constituent* edits (i.e. insertion, deletion, substitution, reorder). Further discussion of structure changes in §B, with examples of structural change sub-types used for manual inspection in Table 5.

**Sentence Split.** A sub-type of a structural edit. We automatically identify split changes prior to annotation, but annotators must first select constituent spans and then associate those spans with the corresponding sentence split. We find the importance of this edit is highly domain-dependent (Figure 13).

### A.1.3 Lexical Simplification

**Paraphrase.** Swapping complex spans with equivalent, simpler alternatives, is the most primitive, yet important, approach to simplification (Qiang et al., 2020) (also referred to as a hypernym, e.g. Štajner, 2016). These are exclusively defined by substitutions marked as *same information* and *positive impact*.

**Trivial Change.** Captures any minor modifications to wording, either through a synonym replacement, or inconsequential change in wording (e.g. the, a). Trivial changes are identified as *trivial insertion*, *trivial deletion* or *trivial substitution*. These edits differ from a content or syntax modification in that they adds no new or major modification to the presentation of information. However, Meister et al. (2020) exemplifies trivial changes should not be ignored as they may modify the information density and verbosity of a sentence. An example is famously shown by Jaeger and Levy (2006):

> EXAMPLE
> How big is the family you cook for?
> How big is the family that you cook for?

The relativizer '*that*' creates no syntactic or conceptual simplicity, but adds clarity as to the identify of the subject. Trivial changes have previously been described with finer granularity, including subcategories like *abbreviation*, *filler words*, *compound segmentation*, *anaphora* (Stodden and Kallmeyer, 2022) or even changes in number/date formatting (Cardon et al., 2022) but we exclude these groups due to their sparsity and our focus on evaluating performance.

## A.2 Error Evaluation

We describe the SALSA error typology, with examples of each type in Table 4. Although despite their sparsity, errors have a far greater impact on fluency and adequacy than individual quality edits (Chen et al., 2023). We refined our definition of errors by focusing on minimizing the amount of error types while retaining the ability to capture the full possibility of simplification ablations. Notably, we specifically exclude a *hallucination* due to its ambiguous definition in related work (Ji et al., 2023), and instead define our error categories to capture any possible hallucination.

### A.2.1 Conceptual Errors

We identify six types of errors in content, with errors primarily being related to information insertion.

**Bad deletion.** As the overwhelmingly most common error, a bad deletion removes necessary and relevant content to the *main idea* of the sentence. As discussed in §A.1.1, the threshold for 'relevancy' is ambiguous.

**Coreference.** More precisely a failure in coreference or anaphora resolution (Maddela et al., 2021), this determines whether an explicit entity reference is removed. This error is only observed on a deletion of information.

> EXAMPLE
> Herbert Spencer's book makes the first...
> His book makes the first...

**Repetition.** Some trivially additional information which simply repeats knowledge already previously contained in the candidate sentence.

> EXAMPLE
> ... the New York City Police Department is a law enforcement agency ...
> ... the New York City Police Department is a police department ...

Despite successfully paraphrasing, *police department*, simply copies content from earlier in the sentence, instead of generating unique information.

**Contradiction.** A negation of the meaning of the original sentence. This notably includes modifying an existing phrase to contradict the original sentence:

> EXAMPLE
> ... the Watergate burglars were convicted ...
> ... the Watergate burglars were not convicted ...

or generating new information making the sentence contradict itself:

> EXAMPLE
> Dextrose adds flavor and texture to dishes, although its consumption is known for negative consequences.
> Dextrose adds flavor, texture and nutrition to dishes, although its consumption is known for negative consequences.

**Factual Error.** We asked annotators to use their commonsense knowledge and limited research to evaluate factuality in edits. Unlike *contradiction*, these claims introduce information which must be externally verified beyond the sentence context. Although factual content is an established focus for summarization evaluation (Pagnoni et al., 2021; Maynez et al., 2020), adequately retaining information (i.e. minimizing *bad deletion*) is a far greater concern for simplification (Devaraj et al., 2022).

> EXAMPLE
> Hilary Clinton was born in 1947.

Hilary Clinton was born in 1947 outside the United States.

In the context of work studying hallucination in LLMs, our *contradiction* and *factual error* categories can be interpreted as intrinsic and extrinsic hallucination respectively (Ji et al., 2023).

**Irrelevant.** A sub-type of a hallucination failing to insert information related to the main idea of the sentence, recognizing the threshold for 'relevancy' is ambiguous (§A.1.1). For simplicity, we report irrelevancy alongside hallucination, as information insertion is generally a rare technique.

### A.2.2 Syntactic Errors

Because syntactic edits are identified by the impact of information distribution, they do not need a fine-grained error typology like conceptual edits, which make a diverse set of modifications. We simply observe each type as a failed attempted at their respective transformations.

**Bad Reorder.** Uses the same word-/phrase-level specification as quality reorder. We also observe that phrase-level reorder errors are almost exclusively observed to introduce a discontinuity to the syntax tree structure (Paetzold and Specia, 2013).

**Bad Structure.** We manually inspect structural errors according to the same sub-type specification as quality edits (§B).

**Bad Sentence Split.** Although sentence splitting is rarely rated as unhelpful, simplifications may unnecessarily segment ideas, or interrupt the flow of information.

### A.2.3 Lexical Errors

Unrelated to information change, lexical errors evaluate primitive issues in fluency or wording.

**Complex Wording.** An attempted paraphrase where the exact meaning is retained, but the replacement uses *more* complex semantics (also referred to as a hyponym, e.g. Stodden and Kallmeyer, 2022).

> **EXAMPLE**
> The researchers conducted an investigation.
>
> The researchers conducted an assay.

**Information Rewrite.** Some substituted span whose content concerns the same *subject*, but fails to substitute the wording correctly, either through misrepresenting or falsely interpreting the information. Although similar to a combination of information deletion and information insertion, the edit is still attempting to represent the same content.

**Grammar Error.** The edit violates grammatical convention. Past error analysis combines *fluency* and *grammar* into the same error type (Maddela et al., 2021) as the two are interrelated. Grammar errors are unique as they can co-occur with other errors, or occur alongside a high quality edit, as sentence fluency is independent from adequacy (Siddharthan, 2014).

### A.3 Edit Severity / Efficacy Levels

We provide examples of each severity level, which are also included as part of annotator training:

> **EXAMPLE**       *Severity: 1 - minor*
> Like so many hyped books before it *The Midnight Library* excited me and gave me pause
>
> *The Midnight Library* excited me and gave me pause

The introductory clause 'Like so many hyped books before it,' situates the sentence within the context of 'hyped books.' However, it does not relate to the main idea of the sentence (the author's opinion on 'The Midnight Library').

> **EXAMPLE**      *Severity: 2 - somewhat*
> Two security flaws, dubbed Meltdown and Spectre by researchers, were made public on 29 January 2018.
>
> Two security flaws, dubbed Meltdown and Spectre by researchers, were made public.

Although the sentence retains its core meaning without 'on 29 January 2018', the specific reference of when 'Meltdown' and 'Spectre' were 'made public' is lost.

> **EXAMPLE**       *Severity: 3 - major*
> If glycolysis evolved relatively late, it likely would not be as universal in organisms as it is.
>
> It likely would not be as universal in organisms as it is.

Since the entity 'glycolysis' has been deleted, the coreference corresponding to the subject 'it' is lost.

### A.4 Overall simplification score

Similar to MQM (Lommel et al., 2014), we collapse edit annotations into a simplification score to allow for direct system comparison. We calculate the sentence-level score as a weighted sum of edit ratings:

$$\sum_{e \in E} \exp\left(\frac{len(e_\mathsf{C}) + len(e_\mathsf{S})}{len(C) + len(S)}\right) \cdot w(e) \cdot r(e)$$

where $S$ is the simplification of complex sentence $C$, $E$ is the set of edits, $e_C$ and $e_S$ are the parts of edit $e$ performed on $C$ and $S$ respectively, $w(e)$ is the edit weight, $r(e)$ is the edit rating (severity /

| Sub-type | Definition | Examples | Original | Simplification |
|---|---|---|---|---|
| Voice Change | Change the subject & receiver of an action | active voice → passive voice | Her book makes the first thorough analysis of this rural society. | The first thorough analysis of this rural society is made by her book. |
| | | passive voice → active voice | Elevation is not primarily considered by the system. | The system does not primarily consider elevation. |
| Part-of-Speech Change | Modifies words' derivation or inflection | nominalisation (verb → noun) | The ability to capture nature scenes has been improving... | The ability to capture nature scenes has seen improvement... |
| | | denominalisation (adjective → verb) | The protesters turned violent when... | The violent protesters... |
| Tense Change | Modifies verb modality or tense | past perfect → past simple | The governor told reporters he had overseen a productive conversation. | The governor oversaw a productive conversation. |
| | | present → past | We compute the Pearson correlation to asses annotation quality. | We computed the Pearson correlation when we assessed annotation quality. |
| Grammatical Number | Distinction of count changes | singular → plural | Victor had scored that goal against the US in 2011, and another in 2012. | Victor had scored those goals in 2011 and 2012. |
| | | generic → specific | The spokesperson for the university called for... | A spokesperson for the university called for... |
| Clausal Change | Modifies predicate structure | coordinate clause → relative clause | Donaldson attempted to speak clearly and he was successful. | Donaldson attempted to speak clearly and successfully. |
| | | subordinate clause → coordinate clause | Although it was raining outside, Jobs continued work in his garage. | Outside it was raining and Jobs continued work in his garage. |

Table 5: Examples of structural modification sub-types used for annotation.

efficacy), and $len$ denotes character length.[4] For weight scheme $w(e)$, we fit a linear regression by considering the sentence-level human ratings gathered in SIMPEVAL$_{2022}$ (Maddela et al., 2023) as a gold standard. As the type of simplification depends on the needs of each particular user group (Stajner, 2021), weights may be adjusted according to the simplification domain (Cemri et al., 2022; Basu et al., 2023; Joseph et al., 2023) or use case (Trienes et al., 2022).

## B Structural Edit Examples

Examples of each structural edit sub-type are listed in Table 5. We find training annotators to label structure change sub-type improved their ability to identify structure changes. We include morphological changes (e.g., *tense change*) as structure edits since these typically require multiple disconnected edits to perform and impact sentence-level meaning. Additionally, other work (Barancikova and Bojar, 2020), specifically Stodden and Kallmeyer (2022) annotate with a larger array of structural changes, notably including separate directions as distinct categories (e.g. singular → plural *and* plural → singular) and including change in sentiment and personal/impersonal form. We exclude these types as they almost never occur in the entirety of the ASSET corpus (Cardon et al., 2022). However,

a case study in Italian simplification (Brunato et al., 2022) shows this structural edit distribution may vary when adapted to the needs of other languages. Similarly, German simplification often converts genitive to dative noun cases, a feature not seen in English simplification (Stodden and Kallmeyer, 2022).

## C Data Collection Details

### C.1 Simplification Systems

Our main corpus of 700 simplifications are from the following diverse simplification approaches:

MUSS (Martin et al., 2022), a BART-large model conditioned on explicit parameter tokens from Martin et al. (2020), fine-tuned on Wiki-Large (Zhang and Lapata, 2017) and mined paraphrase data. MUSS is the SOTA model before GPT-3.5.

T5 (Raffel et al., 2020), an encoder-decoder transformer pre-trained on 745 GB of web text. We use T5-3B and T5-11B variants and fine-tune on the aligned Wiki-Auto dataset (Jiang et al., 2020), shown to be higher quality than Wiki-Large.

GPT-3.5, a series of GPT-3 models pre-trained on text and code dated before Q4 2021. We use the best available text-davinci-003 model, based on InstructGPT (Ouyang et al., 2022), fine-tuned with human demonstrations and reinforcement learning with human feedback. We include both zero- and few-shot (5-shot) generation, using the same

---

[4]We normalize the edit length and use exp to add weight for longer edits.

prompt setup as SIMPEVAL₂₀₂₂ (Maddela et al., 2023).

Humans. We ask two in-house annotators to write simplifications for the 40 newly selected sentences, replicating instructions used in SIMPEVAL₂₀₂₂. We average the annotations of both human simplifications for dataset analysis.

Our test set of 140 simplifications are from recent approaches, including open-source LLMs:

T5 with ACCESS Tokens, we use the same training setup as our fine-tuned T5 model, but prepend the input with ACCESS control tokens (Martin et al., 2020): character length ratio, dependency tree depth ratio, character-level Levenshtein similarity, and inverse frequency ratio. During inference, we use 0.9 for the length ratio, and 0.75 for the other three control tokens, following the setup in (Maddela et al., 2023).

Alpaca-7B (Taori et al., 2023), a fine-tuned LLaMA model (Touvron et al., 2023) on 52K GPT-3.5 outputs generated using the Self-Instruct technique (Wang et al., 2023). As we find the prompt used for GPT-3.5 is too complex for Alpaca, we use the following prompt:

*"Rewrite the following complex sentence in order to make it easier to understand by non-native speakers of English."*

Vicuna-7B (Chiang et al., 2023), a fine-tuned LLaMA model on 70K publicly shared ChatGPT conversations. As the training data for Vicuna includes prompts that are more diverse and complex than those used by Alpaca, Vicuna can manage longer prompts, but not at the level of GPT-3.5, so we use the following prompt:

*"Rewrite the following complex sentence in order to make it easier to understand by non-native speakers of English. The final simplified sentence needs to be grammatical, fluent, and retain the main ideas of its original counterpart without altering its meaning."*

ChatGPT, an optimized chat variant of GPT-3.5, the model we use is gpt-3.5-turbo-0301.

GPT-4, a large multimodal model that performs better than GPT-3.5 models. We use the version of gpt-4-0314.

For ChatGPT and GPT-4, we use the same prompt as GPT-3.5:

*"Rewrite the following complex sentence in order to make it easier to understand by non-native speakers of English. You can do so by replacing*

|  | Fleiss kappa ($\kappa$) | ⅔ Agree% | % sentences |
|---|---|---|---|
| Bad Deletion | 0.51 | 64 | 35 |
| Complex Wording | 0.26 | 32 | 20 |
| Information Rewrite | 0.27 | 26 | 10 |
| Grammar Error | 0.17 | 18 | 10 |
| Bad Structure | 0.02 | 6 | 10 |
| Bad Reorder | 0.14 | 19 | 9 |
| Irrelevant | 0.22 | 26 | 8 |
| Bad Split | 0.13 | 17 | 4 |
| Repetition | 0.33 | 30 | 4 |
| Contradiction | 0.19 | 25 | 1 |
| Coreference | 0 | 0 | 0 |

Table 6: Fleiss kappa error identification agreement measured per-sentence alongside error frequencies. As errors were far more rare, we observe a strong relationship between frequency and expected agreement.

*complex words with simpler synonyms (i.e. paraphrasing), deleting unimportant information (i.e. compression), and/or splitting a long complex sentence into several simpler ones. The final simplified sentence needs to be grammatical, fluent, and retain the main ideas of its original counterpart without altering its meaning."*

Humans. As existing automatic simplification evaluation metrics rely on human references, we include two human-written simplifications to use for metric evaluation, but do not collect annotations on these references.

## C.2 Interpreting Annotator Agreement

As the SIMPEVAL challenge dataset contains more edits than past simplification corpora, edit annotation becomes significantly more challenging as multiple groups of edits often overlap and simplifications contain more compression and sentence-level transformations. Additionally, error-prone systems like MUSS make it challenging to disambiguate error and quality edits. Figure 8 illustrates an example of this disagreement, showing many of the same tokens are annotated, but with different edit spans. For example, observe the last clause in the sentence, which performs a rewrite:

> **EXAMPLE**
> *that the fort stood out for its defenders' heroic resistance.*
> *and the defenders of the fort gave their lives to save the city.*

We see three different, but valid understandings of this phrase:

1. Information was replaced - The information about the defenders' *resistance* is inherently different then the defenders' giving *their lives to save the city* and is therefore an add/deletion pair.

Original Sentence (Human Written):
Repeatedly besieged during the War of Restoration of Independence, it was during the siege of the city of Elvas, led by the Spanish commander Don Luis de Haro in 1658, that the fort stood out for its defenders' heroic resistance.

Simplified Sentence (Human or AI Model Written):
It was often attacked during the War of Restoration of Independence. In 1658, the Spanish commander Don Luis de Haro attacked the city of Elvas, and the defenders of the fort gave their lives to save the city.

Original Sentence (Human Written):
Repeatedly besieged during the War of Restoration of Independence, it was during the siege of the city of Elvas, led by the Spanish commander Don Luis de Haro in 1658, that the fort stood out for its defenders' heroic resistance.

Simplified Sentence (Human or AI Model Written):
It was often attacked during the War of Restoration of Independence. In 1658, the Spanish commander Don Luis de Haro attacked the city of Elvas, and the defenders of the fort gave their lives to save the city.

Original Sentence (Human Written):
Repeatedly besieged during the War of Restoration of Independence, it was during the siege of the city of Elvas, led by the Spanish commander Don Luis de Haro in 1658, that the fort stood out for its defenders' heroic resistance.

Simplified Sentence (Human or AI Model Written):
It was often attacked during the War of Restoration of Independence. In 1658, the Spanish commander Don Luis de Haro attacked the city of Elvas, and the defenders of the fort gave their lives to save the city.

Figure 8: Edit selection between three annotators on a MUSS simplification. For complex examples, multiple valid interpretations for span labeling may exist, however we find annotator's overall judgements are consistent.

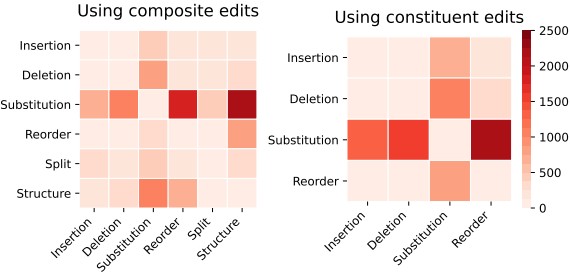

Figure 9: Edit-level confusion of annotated tokens. The vertical and horizontal axes represents the class of majority agreement and minority decision between annotators respectively. The left includes all edits, while the right calculates agreement using the underlying constituent spans selected for structure and split edits.

2. Information was retained, but paraphrased - The phrase *heroic resistance* being equivalent in meaning to *gave their lives*.
3. Subject was modified and information was replaced - The subject swap between the subject of the clause being the *fort* to being the *defenders*. The rest being an add/deletion pair.

Varying interpretations of the same edit leads to natural disagreement. However, often a clear annotation exists and is not captured. For example, although we instructed annotators to create *separate* edits for overlapping syntax and conceptual edits, this occurred inconsistently in practice:

> **EXAMPLE**
> it was during *the siege of* the city of Elvas
> Don Luis de Haro *attacked* the city of Elvas

1. Identified the edit as a structural change, because the noun *siege* was replaced with a verb, modifying the voice of the sentence
2. Identified a paraphrase, annotating *siege* as a more complex word than *attacked*
3. Correctly identified both edits occurred simultaneously

We find the largest source of disagreement comes from overlapping edits of multiple types, most often between structural changes and other types, because they often co-occur. Figure 9 demonstrates structural edits explain a significant portion of disagreement. Additionally, because structural edits are a *composite* edit, the same spans are captured by the structural edits' constituent spans and recalculating agreement using these spans, disagreement instead focuses on whether tokens are substituted.

Within individual sentences, we often observe multiple valid interpretations for span labeling, highlighting the inherit ambiguity in the task. Despite this, annotators still successfully communicated edit performance. All three annotators identified both the *bad deletion* and *hallucination* errors contained in the sentence. For the full SIMPEVAL dataset, we report error identification agreement in Table 6, finding syntax errors (e.g., *bad structure*, *bad reorder*) are far more difficult to identify than content or lexical errors. Particularly, *complex wording* and *grammar* errors exhibit both high fre-

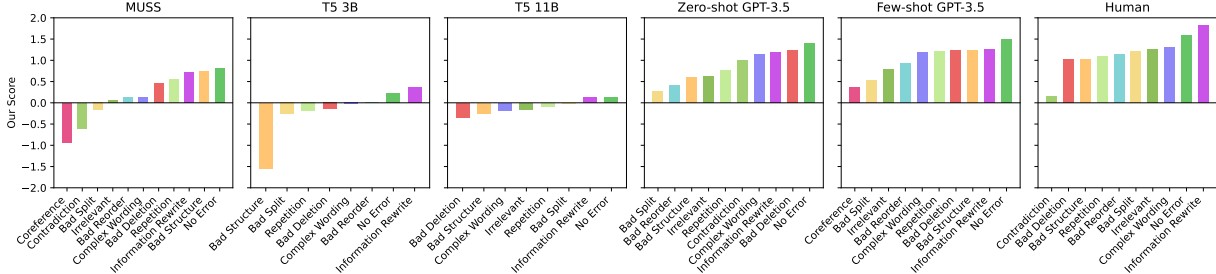

Figure 10: Average sentence-level score across error sentences for each system.

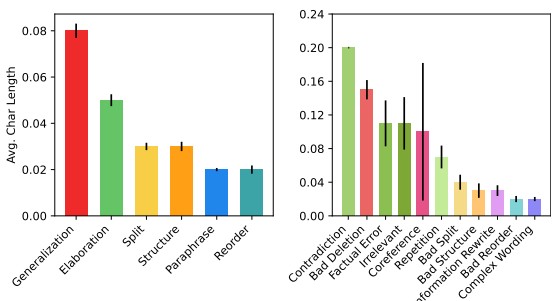

Figure 11: Average edit coverage of edit types and specific error types with 95% confidence interval. Edit coverage, the ratio of the simplification being edited, is formalized in §A.4.

quency and high agreement, as the definitions of these errors are unambiguous. Broadly, we find that high span-level agreement is not necessary for capturing overall, or even fine-grained sentence-level performance, a clear trade-off exists between the granularity of annotations and expected agreement.

## D  Further Analysis

Here, we report additional findings on the SIM-PEVAL dataset and model performance, alongside observations about edit-level evaluation as a task. Figure 11 reports the average edit coverage by each edit operation and error type. We find paraphrases are typically annotated as pairs of a few words, while conceptual edits typically occur on the clause level and are annotated together. Surprisingly, structure changes often occurred as a few words:

> **EXAMPLE**                                              *MUSS*
> ... Corbin has expanded his business to include agri-tourism, using his farm to host weddings ...
> ... Corbin's business also offers agritourism and he uses his farm to host weddings ...

The edit converts the beginning subordinate clause to a coordinate clause, yet only requires substituting a single word. Errors exhibited a significantly higher variance in size, which may be attributed to their sparsity, as no error except *bad deletion* occurs

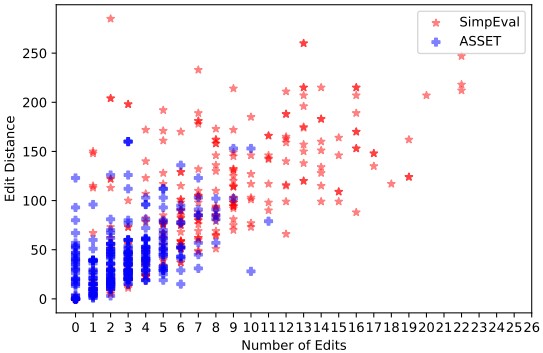

Figure 12: Edit distance and number of annotated edits for 300 randomly sampled sentences from ASSET and SIMPEVAL. While past work found no relationship, by extending ASSET to more complex sentences we see a clear correlation arise.

in more than 20% of outputs (Table 6). However, error sizes display the same trend as their quality counterparts, with conceptual errors typically being seen on the clause level. We also found single-word conceptual errors such as:

> **EXAMPLE**                                    *Zero-shot GPT-3.5*
> ... Arroyo released a statement that acted as an informal concession of sorts ...
> ... Arroyo released a statement that was like a formal concession.

> **EXAMPLE**                                    *Few-shot GPT-3.5*
> The sentence included a fine of $400...
> They imposed a fine of $400...

Were less frequent than hallucinating entirely new phrasing or ideas. This may be promising for error detection as it implies error spans are often clausal and occur among many adjacent tokens.

**Quality and Error Are Interrelated.** Figure 10 displays sentence-level scores for our error typology across systems on SIMPEVAL. We find the existence of an error to be a consistent predictor of a lower quality sentence, even in human simplifications. However, we find some errors correlate with a higher score (e.g. bad structure, information rewrite), but this may be attributed to the multi-clause complex sentences in SIMPEVAL having a

|  | Lexical | | Syntax | | Conceptual | | Error | | Quality | | Overall | |
|---|---|---|---|---|---|---|---|---|---|---|---|---|
|  | $\mu$ | $\sigma$ | $\mu$ | $\sigma$ | $\mu$ | $\sigma$ | $\mu$ | $\sigma$ | $\mu$ | $\sigma$ | $\mu$ | $\sigma$ |
| MUSS | 0.81 | 1.23 | 0.45 | 0.64 | 0.97 | 1.73 | 0.66 | 1.03 | 1.00 | 1.04 | 1.77 | 1.66 |
| T5 3B | 0.24 | 0.56 | 0.18 | 0.38 | 0.65 | 1.92 | 0.34 | 0.97 | 0.39 | 0.62 | 0.76 | 1.15 |
| T5 11B | 0.22 | 0.44 | 0.17 | 0.92 | 0.61 | 1.78 | 0.36 | 1.30 | 0.32 | 0.71 | 0.71 | 1.51 |
| Zero-shot GPT-3.5 | 1.32 | 1.40 | 0.67 | 0.65 | 0.17 | 0.38 | 0.34 | 0.53 | 1.75 | 1.55 | 2.10 | 1.60 |
| Few-shot GPT-3.5 | 1.41 | 1.43 | 0.57 | 0.53 | 0.15 | 0.39 | 0.25 | 0.46 | 1.80 | 1.49 | 2.11 | 1.60 |
| Human | 1.25 | 1.85 | 0.60 | 0.86 | 0.32 | 0.83 | 0.25 | 0.62 | 1.67 | 1.64 | 2.04 | 2.16 |

Table 7: Mean ($\mu$) and std. deviation ($\sigma$) of average sentence-level SALSA sub-scores across systems. Human simplification may be interpreted as highly simplified ($\mu = 2.04$) and highly diverse ($\sigma = 2.16$).

a far greater number of positive edits when these corresponding errors occur. Broadly, we observe an inverse relationship between error and quality. As the error score increases (a function of the severity, frequency and size of errors), the quality must decrease.

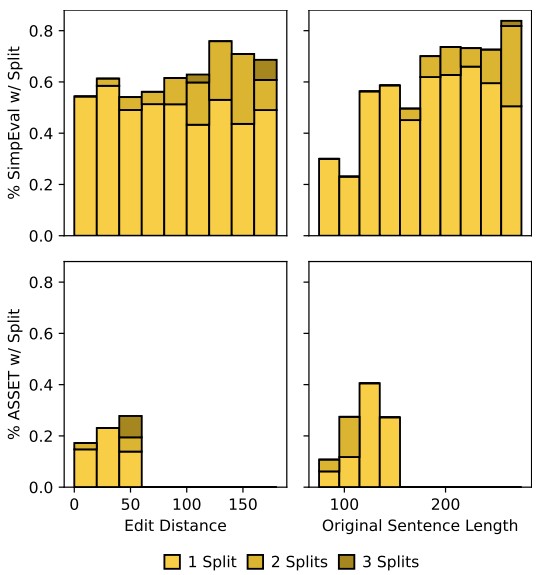

Figure 13: Proportion of sentences containing at least a single split. Although ASSET has a much lower frequency of sentence splits (32%), a longer input sentence implies a sentence split is more likely to occur.

**Increased Edits Enables, But Does Not Guarantee Performance.** Table 7 reports the mean and variance of sub-scores for the sentence-level SALSA score across each system. Edit-level scoring addresses the frequent evaluation concern that conservative systems may maximize their score by performing a minimal number of *safe* edits (Alva-Manchego et al., 2021). The qualitatively conservative simplifications of T5 and zero-shot GPT-3.5 often score low because they fail to make many edits. SALSA distinguishes the MUSS simplifications with many successes, but more failures than other systems. We find the extent of sentence editing is not heuristic, but is a prerequisite for high

performance and that overall simplification performance is often determined by a small number of high-impact edits.

**Sentence Length Impacts Edit Frequency.** Previous linguistic annotation of the ASSET corpus (Cardon et al., 2022) reports that the number of modifications to a sentence does not correlate with input size. In Figure 13, we observe the same relationship on ASSET, however – because ASSET only represents simplifications of simpler sentences typically containing a single idea – when we extend the analysis to the more complex SIMPEVAL dataset, we see a clear relationship between the edit distance and the number of transformations in simplifications across all systems. This is also best exemplified by the split edit, which often signifies too many ideas are being contained within a single sentence. Figure 12 demonstrates the proportion of simplifications which exhibit a split across sentence lengths and edit distance. While split edits within ASSET were generally low, the much longer SIMPEVAL simplifications almost guaranteed all systems performed a sentence split. These findings highlight that performance measures should be length-agnostic, as to guarantee simplifications which simply contain more transformations due to a longer original sentence length are not arbitrarily rated as higher quality.

**Composite Edits.** We report the breakdown of constituent edits in structure and split edits in Figure 15. Split edits typically need to rewrite the conjunction through inserting & deleting discourse tokens, while structure edits are typically performed some syntax *transformation* to the existing sentence tree, more often requiring substituted or reordered tokens.

**SALSA Test Set.** Figure 16 reports the frequency of quality and error edits on the novel SALSA test set systems. While adding control tokens to T5 substantially improves the frequency of edits, we

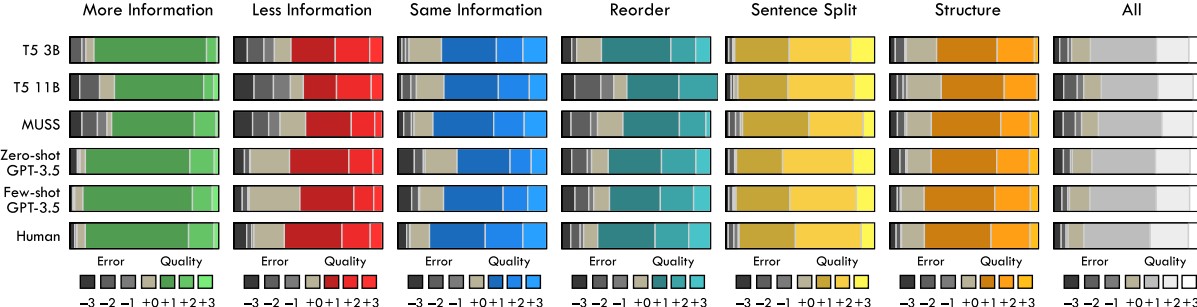

Figure 14: Distribution of ratings of edits on SIMPEVAL per-model. Compared to edit coverage distributions (Fig. 6), we see the same underlying relationship, but the difference in error and GPT vs. human quality is less exaggerated. This figure does not reflect the typically much longer human simplification spans and fine-tuned models' error spans.

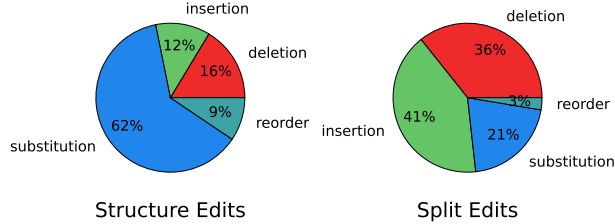

Figure 15: Breakdown of composite edits by the % makeup of their constituent tokens.

|  | Edit Type | # Edits | # Tokens | Avg. Rating |
|---|---|---|---|---|
| Quality Evaluation | Elaboration | 1947 | 4996 | 0.25 |
| | Generalization | 2644 | 10650 | 0.52 |
| | Word Reorder | 67 | 744 | 0.55 |
| | Component Reorder | 823 | 7273 | 0.62 |
| | Sentence Split | 1605 | 6759 | 0.71 |
| | Structure Change | 2013 | 10614 | 0.48 |
| | Paraphrase | 4394 | 18278 | 0.7 |
| Error Evaluation | Bad Deletion | 771 | 5019 | -0.66 |
| | Coreference | 9 | 42 | -0.89 |
| | Repetition | 107 | 572 | -0.53 |
| | Contradiction | 4 | 22 | -1.75 |
| | Factual Error | 75 | 421 | -1.15 |
| | Irrelevant | 64 | 347 | -0.67 |
| | Bad Word Reorder | 18 | 160 | -0.61 |
| | Bad Component Reorder | 243 | 2512 | -0.81 |
| | Bad Structure Change | 291 | 1808 | -0.49 |
| | Bad Sentence Split | 143 | 749 | -0.94 |
| | Complex Wording | 597 | 2412 | -0.45 |
| | Information Rewrite | 159 | 877 | -0.65 |
| | Grammar Error | 71 | 372 | -0.96 |
| | Trivial Change | 2983 | 8876 | 0 |

Table 8: Basic collected statistics on our edit-level SALSA annotations of SIMPEVAL. # tokens indicates tokens highlighted within each edit's spans.

find it still underperforms both MUSS and LLMs. Additionally, T5-11B makes a surprising increase in error frequency relative to the increase in the number of edits it performs relative to T5-3B. Language models demonstrate a smooth increase in edits, with the exception of GPT-4 making significantly less conceptual edits. Manual analysis reveals its conceptual edits are often sentence-level operations, which are not reflected in edit counts. The LLaMA-based Alpaca and Vicuna demonstrate surprisingly strong performance despite their relatively small size and training setup, even outperforming the fine-tuned simplification models.

**SALSA Dataset Statistics.** We report full statistics on all 840 simplifications in Table 8. Similar to FRANK (Pagnoni et al., 2021), we asked annotators to note edits that could not be annotated, and we observe less than 0.5% of edits were not captured by one of our edit types. We consider the SALSA framework *complete*.

## E    Further Word-level QE Results

We include test set word-level F1 score on words in the original sentence, simplified sentence, and both sentences (same as Table 3) in Table 9. In the original sentence, only deletion edits are labeled. Thus, the performance in the original sentence column indicates the model's ability to identify quality or error deletion edits. The best-performing method, Ec-One, achieves over 50% in both quality and error F1. For the simplified sentence, which contains substitution and insertion edits, the model delivers better quality F1 but experiences a drop in error F1. This could be due to the higher proportion of error edits in deletion compared to substitution and insertion. In addition, the edit classification approach significantly improves the error F1 on the simplified sentence, compared to the tagging approaches, which reflects that tagging methods fail to capture multiple types of edits and those spanning both sentences like substitutions.

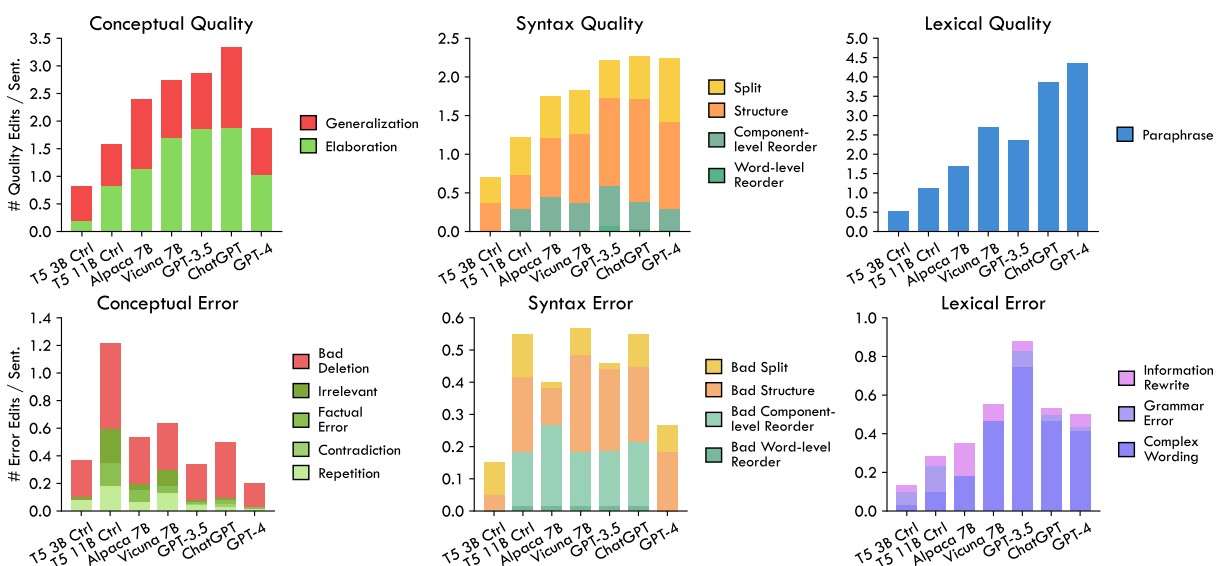

Figure 16: Overview of success and failure edits on the SIMPEVAL test set, collected using held-out simplification models as detailed in §C.1.

| Method | Original Sentence | | | Simplified Sentence | | | Overall | | |
|---|---|---|---|---|---|---|---|---|---|
| | Quality | Error | Ok | Quality | Error | Ok | Quality | Error | Ok |
| *End-to-end* | | | | | | | | | |
| Tag | 36.53 | 52.20 | 94.79 | 73.83 | 13.85 | 90.18 | 67.00 | 28.24 | 92.88 |
| Tag-ML | 42.17 | 46.88 | **95.22** | **76.86** | 20.61 | 90.03 | 70.73 | 30.06 | **93.09** |
| *Two-stage (use word aligner to get edit information)* | | | | | | | | | |
| Tag-EI | 33.06 | 46.04 | 94.76 | 76.75 | 20.00 | **90.66** | **69.09** | 30.37 | 93.04 |
| Ec-Sep | 45.70 | 49.18 | 93.19 | 71.50 | 25.66 | 89.43 | 64.87 | 36.15 | 91.56 |
| Ec-One | **50.53** | **54.63** | 93.30 | 74.97 | **28.15** | 90.04 | 68.77 | **39.50** | 91.91 |
| Oracle (Ec-One) | 84.51 | 75.88 | 99.33 | 89.45 | 65.26 | 97.03 | 88.31 | 69.44 | 98.35 |

Table 9: Word-level F1 scores of different methods on SALSA test set, organized by the edit belonging to the original or simplified sentence.

## F  Implementation Details

### F.1  Generating Simplifications (§3.1)

For all prompted models, we follow the hyperparameters of SIMPEVAL$_{2022}$ (Maddela et al., 2023), using temperature=1.0 and top-p=0.95. For all T5 variants, we train them on the Wiki-Auto corpus (Jiang et al., 2020) using 8 A40 GPUs for 8 epochs with a batch size of 64. We use a learning rate of 3e-4 and AdamW (Loshchilov and Hutter, 2019) as the optimizer. For MUSS, we replicate the original setup (Martin et al., 2022). We use beam search with a beam size of 10 for these fine-tuned models.

### F.2  Automatic Metrics (§5)

**Baseline Automatic Metrics.** We use RoBERTa-large as the base model for BERTSCORE and the best available wmt21-comet-mqm as COMET-MQM.

**LENS-SALSA.** Our implementation is based on

the reference-less COMETKIWI metric for machine translation (Rei et al., 2022). We modify their task setup of predicting binary quality labels for each output word $\hat{y}_i \in \{\text{OK}, \text{BAD}\}$ to a regression task using labels $\hat{y}_i \in [-3, 3]$, corresponding to each word rating in their SALSA annotations, as we find it performs better than using binary or three class labels in our preliminary study. Our regression task optimizes MSE loss on the word rating objective, rather than Cross Entropy Loss. The training objective can be formalized as:

$$\mathcal{L}_{sent}(\theta) = \frac{1}{2}(y - \hat{y}(\theta))^2$$

$$\mathcal{L}_{word}(\theta) = -\frac{1}{n}\sum_{i=1}^{n}\frac{1}{2}(y_i - \hat{y}_i(\theta))^2$$

$$\mathcal{L}(\theta) = \lambda_s\mathcal{L}_{sent}(\theta) + \lambda_w\mathcal{L}_{word}(\theta)$$

| | | BLEU | SARI | BERTSCORE | COMET-MQM | LENS | LENS-SALSA |
|---|---|---|---|---|---|---|---|
| Quality | Lexical | -0.185 | 0.030 | 0.015 | 0.086 | **0.289** | 0.284 |
| | Syntax | -0.117 | 0.097 | 0.008 | 0.024 | 0.206 | **0.244** |
| | Conceptual | -0.240 | -0.147 | -0.325 | -0.187 | -0.006 | **0.173** |
| Error | Lexical | -0.259 | -0.162 | -0.134 | -0.004 | -0.059 | **0.015** |
| | Syntax | -0.147 | -0.094 | -0.136 | -0.073 | -0.042 | **-0.013** |
| | Conceptual | -0.128 | -0.099 | -0.293 | -0.169 | -0.016 | **0.062** |
| All | All Error | -0.263 | -0.190 | -0.329 | -0.170 | -0.035 | **0.046** |
| | All Quality | -0.201 | 0.056 | -0.018 | 0.033 | 0.304 | **0.318** |
| | All Edits | -0.286 | -0.035 | -0.235 | -0.129 | 0.266 | **0.336** |

Table 10: Pearson correlation between automatic metrics and SALSA sub-scores on the validation set, with test set performance reported in Table 2.

where $\lambda_s$ and $\lambda_w$ weight word- and sentence-level losses. We experimented with custom weighting for edit ratings, but did not fine performance improvements. For fine-tuning, we set $\lambda_w = 0.9$.

The COMETKIWI design aggregates hidden states using a scalar mix module, and uses two feed forward networks for sentence- and word-level training. For pre-training, we optimize a RoBERTa-large model on the sentence-level SIMPEVAL training data used to train LENS (Maddela et al., 2023), with the training setup using only a single MSE loss to predict the sentence-level score (i.e., $\lambda_s = 1$, $\lambda_w = 0$). We follow COMETKIWI and freeze parameter updates for the RoBERTa encoder for the first epoch and use a learning rate of 1e-5 and 3e-5 for pre-training and fine-tuning respectively. We pre-train and fine-tune for 5 epochs, using the model with the highest validation set performance. We report the corresponding validation performance in Table 10.

### F.3 Edit Classification (§6).

All experiments are conducted using 2 A40 GPUs. We use the AdamW optimizer with a weight decay = 0.01, and implement our models using the Hugging Face Transformers. Learning rate are swept over 1e-5, 2e-5, 5e-5, 8e-5 for each method. Each run is trained for eight epochs with a batch of 32. This results in training times of less than five minutes per run for tagging methods and less than 20 minutes per run for the edit classification methods. We perform an evaluation of the validation set at each training step and use the model that achieved the highest validation performance on the test set.

For the word alignment model used in the two-stage approach, we adopt the QA-based word aligner (Nagata et al., 2020), which formulates the task in a SQUAD style (Rajpurkar et al., 2018). We use RoBERTa-Large as the base model. We first pre-train it on monolingual word alignment datasets MultiMWA-Wiki and MultiMWA-Newsela from (Lan et al., 2021), and then fine-tune it on the SALSA annotations in the training set. During both pre-training and fine-tuning stages, we perform a learning rate sweep over {1e-5, 2e-5, 5e-5, 8e-5} and train for 5 epochs, and save checkpoint at the end of every epoch. The highest evaluated checkpoint (pre-train for 2 epochs and fine-tune for 2 epochs) is selected for testing, achieving 81.03 F1 on the validation set.

On a side note, for the word that is tokenized into multiple tokens, we use its first token for prediction.

## G Annotation Tutorial

We include screenshots to highlight the diversity of exercises and interactive elements in our detailed interface tutorial.

# Text Simplification Annotation Tutorial

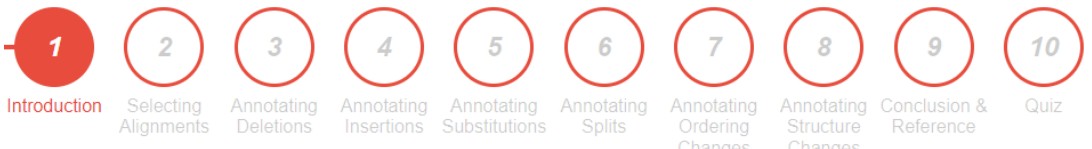

# Introduction

Welcome! In this project you will read simplified sentences generated by Artificial Intelligence and rate their quality.

This qualification HIT will train you to perform this task. You must be able to:

- Find the changes our AI made (i.e. "**selecting spans**")
- Evaluate the quality of each change
- Identify errors in each change

Here's a big picture of what we're doing:

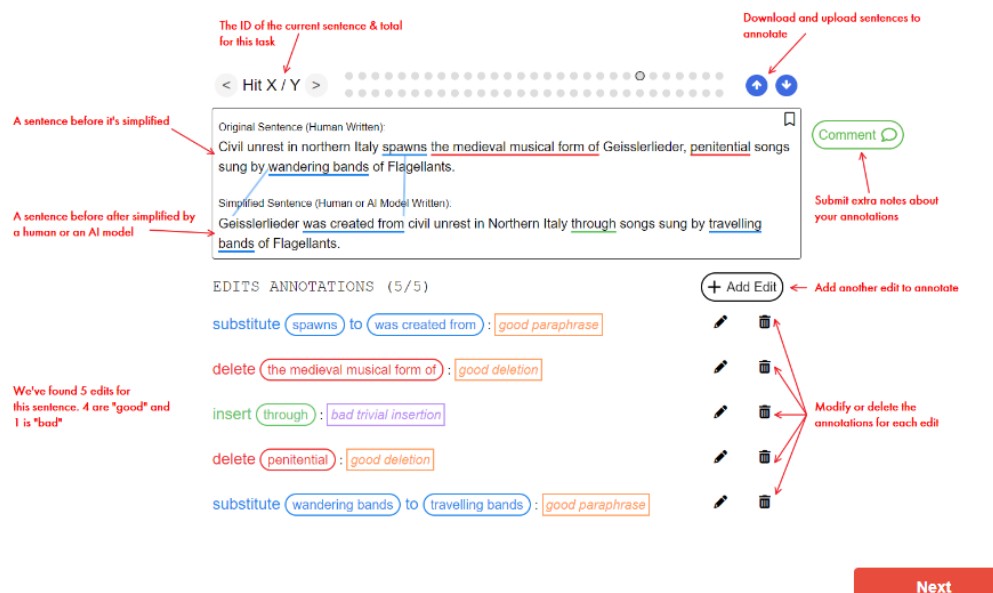

Figure 17: Landing page introducing annotators to each part of the task. The 10 stages organize different concepts in the SALSA typology.

## Examples

Observe this original sentence:

> Original Sentence (Human Written):
>
> Born into slavery in Virginia in 1856, Booker T. Washington became an influential African American leader at the outset of the Progressive Era.
>
> Simplified Sentence (Human or AI Model Written):
>
> Booker T. Washington became an influential African American leader at the outset of the Progressive Era.

This sentence communicates many different facts. Here are just a few:

- Booker T. Washington was born in Virginia
- Booker T. Washington was born in 1856
- Booker T. Washington was born into slavery
- Booker T. Washington was an influential leader
- Booker T. Washington was a African American leader
- Booker T. Washington was a influential African American leader as a result of being born into slavery
- Booker T. Washington was a leader at the beginning of the Progressive Era

Hover over each piece of information to see which part of the sentence could be deleted to remove that information from the sentence. As you can see, sentences which communicate many ideas may be hard to narrow down whether a span is *significant*.

In this case, the *main idea* of the sentence is *Booker T. Washington is an influential African American leader.* Deletions are necessary for text simplification, we just want to ensure this *main idea* is still being communicated.

Let's put it together with a few other sentences:

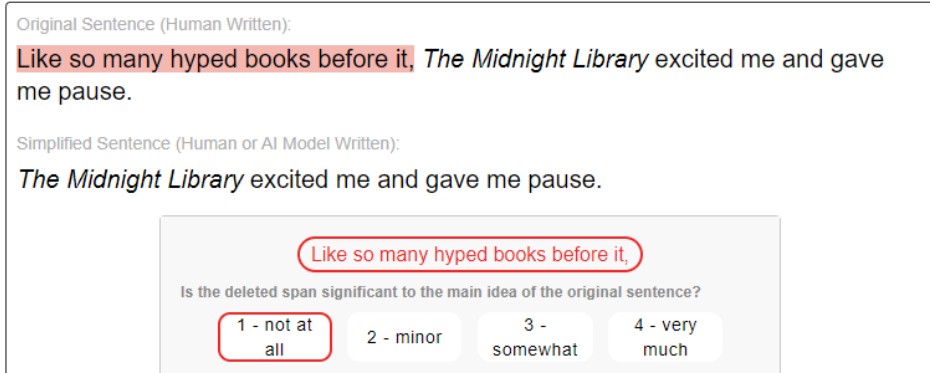

Figure 18: Example interactive allowing annotators to see different spans to understand different amounts of relevancy to the *main idea* of the sentence.

## Phrase vs. Syntax Edits

As you can see *phrase edits* can only capture how individual pieces of information or a small set of words change. When the AI creates an edit which modifies the sentence as a whole, this *must* be captured by a structural edit. Here's some examples of overlapping phrase and syntax edits:

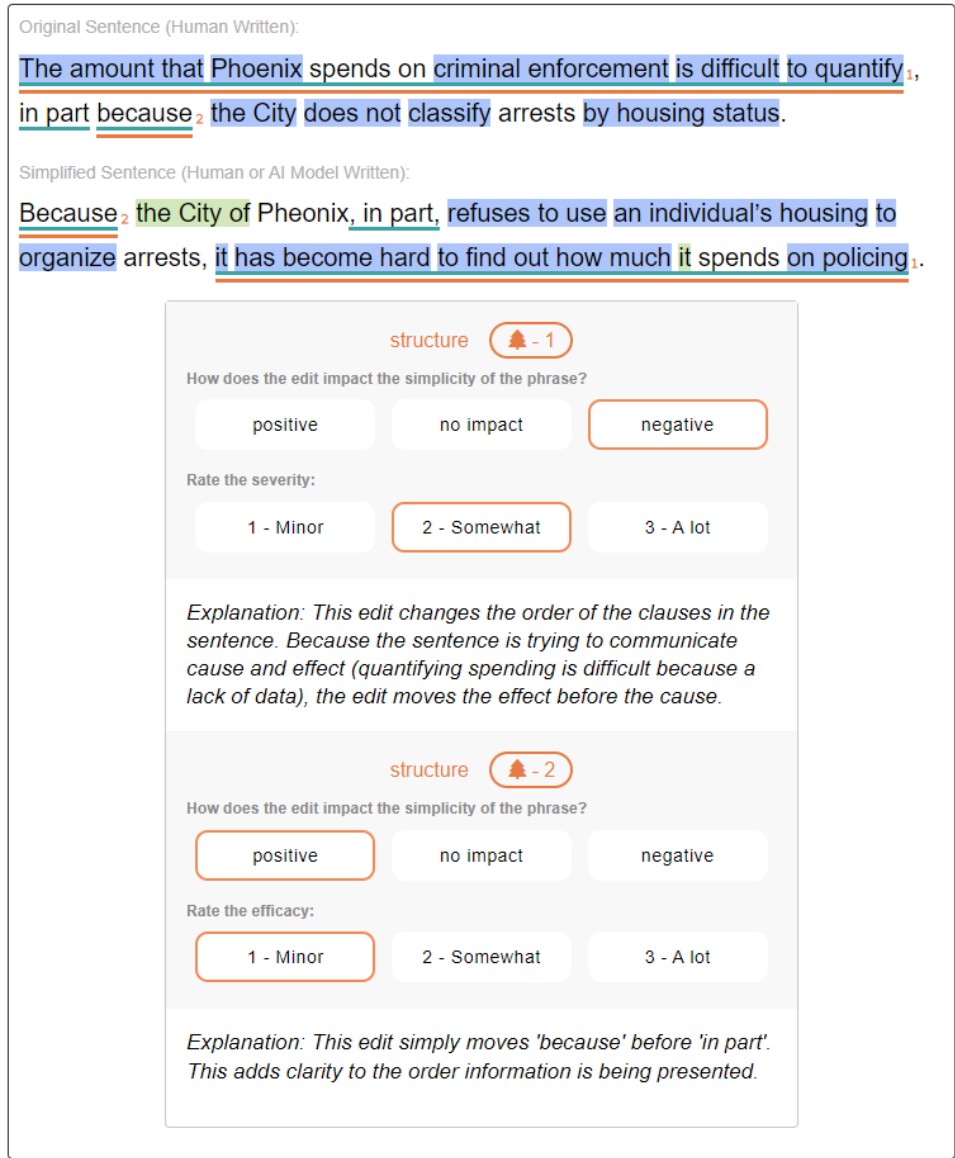

Figure 19: One of the 100 sentence examples provided to annotators, highlighting different types of structure edits existing within the same sentence.