# OpenReview forum: "Dancing Between Success and Failure: Edit-level Simplification Evaluation using SALSA"
_EMNLP/2023/Conference — EMNLP 2023 Main_

### Official Review · Reviewer_nj2X · 2023-08-03

**Soundness:** 5

**Excitement:**

4: Strong: This paper deepens the understanding of some phenomenon or lowers the barriers to an existing research direction.

**Missing References:**

A reference-free metric for TS has been proposed in prior work. Please correct L624 and add a citation to Kriz, Reno, Marianna Apidianaki, and Chris Callison-Burch. "Simple-qe: Better automatic quality estimation for text simplification." arXiv preprint arXiv:2012.12382 (2020).

**Paper Topic And Main Contributions:**

The paper proposes a new evaluation framework for evaluating simplified sentences using 21 linguistically grounded edit types. The authors collect 19k high-quality annotations on 840 simplified outputs from multiple systems where the quality is ensured by training sessions of 6 native U.S. undergraduate students. A new word-level quality estimation task with baseline approaches is proposed where the proposed reference-free metric LENS-SAMSA is shown to outperform other TS evaluation metrics. Their findings show that GPT generates more diverse edits than human-based simplification but also introduces more errors compared to human reference simplification.

**Questions For The Authors:**

1. It is unclear why it is claimed in Figure 3 that MUSS performs paraphrasing at a human rate.
2. It might be helpful to clarify whether the MUSS model is supervised or unsupervised in the main text.
3. L188: Unclear whether and how this definition of edit coverage differs from the equation presented in Figure 5 and it is also unclear how to read Figure 5. Could you explain why for example in more information, muss has a higher overall proportion than other models?

**Reasons To Accept:**

1. The paper proposes a new evaluation framework and a dataset as well as a word-level QE task for text simplification which could help significantly advance research in TS evaluation beyond coarse-grained Likert-scale rating.
2. The proposed reference-free metric achieves better correlation with human judgments on both sentence and word-level quality estimation tasks on the collected benchmark,
3. A thorough evaluation is conducted to study how human-based edits differ from model-generated edits.

**Reasons To Reject:**

1. While the evaluation focuses on the objective aspect of what constitutes a successful or a failed edit, the framework still does not give much information about who are these simplifications useful for as what constitutes simplified really depends upon the target audience. This has been acknowledged in L288 but it might be useful to define what is considered a successful or a failed edit as observed in the data or as was considered by the authors more directly.
2. Not a reason for rejection and really appreciate all the hard work put into this work but that also makes it difficult to follow the core contributions, breaking this paper into two would greatly benefit the readers as a lot of interesting and relevant content is currently in the appendix material. For example Figure 7 really helps ground success and failure edits which if it is indeed the core contribution should ideally be in the main text. For word-level QE results, table 9 is in appendix material.

**Reproducibility:**

4: Could mostly reproduce the results, but there may be some variation because of sample variance or minor variations in their interpretation of the protocol or method.

**Reviewer Confidence:**

4: Quite sure. I tried to check the important points carefully. It's unlikely, though conceivable, that I missed something that should affect my ratings.

**Typos Grammar Style And Presentation Improvements:**

Section 6 should come earlier as it helps ground the nature of the document simplified, without this information it was hard to follow the results discussed before.

---

> ### Author Rebuttal · Authors · 2023-08-29
>
> Thank you sincerely for your review and your dedicated time to this.
>
> ---
>
> **Responses to your comments:**
>
> > “It might be useful to define what is considered a successful or a failed edit as observed in the data or as was considered by the authors more directly.”
>
> Thank you for the suggestion. Our definition of success and failure is based on the general notion of simplicity, which is whether an edit makes the sentence simpler or more complex. Our 21 specific categories of successful or failed edits are refined through initial pilot rounds of annotations.
>
> ---
>
> **Answers to your questions:**
>
> > “Why it is claimed in Figure 3 that MUSS performs paraphrasing at a human rate.”
>
> Thank you for pointing this out. This point was intended to contextualize Figure 3 w.r.t Figure 5, which shows the edit coverage for paraphrasing to be similar to humans. We will clarify this statement in our final version.
>
> > “It might be helpful to clarify whether the MUSS model is supervised or unsupervised in the main text.”
>
> We will clarify this in our final version.
>
> > “How the definition of edit coverage at L188 differs with the equation in Figure 5 and how to read Figure 5. Could you explain why for example in more information, muss has a higher overall proportion than other models?
>
> L188 should be _the length of each edit in proportion to the total length of the complex sentence and simplification_ as shown in the equation. We will correct it in the final version.
>
> Regarding Figure 5, each column should be considered independently, as they are rescaled to facilitate comparison between different models.
>
> MUSS having a higher overall proportion in the "more information" category is because it makes longer and more error edits. We observe that MUSS often generates its own interpretations that are not well-grounded in the context.

---

### Official Review · Reviewer_6XdU · 2023-08-05

**Soundness:** 3

**Excitement:**

4: Strong: This paper deepens the understanding of some phenomenon or lowers the barriers to an existing research direction.

**Paper Topic And Main Contributions:**

This paper proposes an annotation framework for evaluation of simplification tasks, called SALSA. Using SALSA, 19K annotations are collected. The authors demonstrate the usefulness of the framework by showing how some data can be used. First, they analyze several simplification models and human edits. Second, they used annotations to evaluate automatic evaluation metrics such as BLEU. Moreover, they propose a new automatic evaluation metric, Lens-SALSA, which is trained on the annotation data. Finally, they show that the data can be used for word-level quality estimation tasks.

**Questions For The Authors:**

Question A: I understand that COMET-MQM is an evaluation metric for translation that takes source, hypothesis, and reference (source and hypothesis or reference are different languages) as inputs and gives a score.  How did you adapt COMET-MQM for the simplification task?

**Reasons To Accept:**

- This paper proposes a useful framework for evaluation for simplification tasks.
- It offers a comprehensive analysis of the collected data by SALSA.
- In addition to analyzing simplification models, they also demonstrate the analysis of automatic evaluation metrics and their applied use in QE tasks.


**Reasons To Reject:**

While this paper shows many ways to use the data collected by the proposed framework, I feel that the framework has not been fully evaluated. For example, I think there needs to be a comparison of the SALSA evaluation process and the process of annotating the 21 edit types directly. Consequently, the evaluation of the framework is also affected by the reliability of the data collected in this study. In fact, some edit types have a low agreement (Table 3), which also affects the reliability of the various analyses based on that data (Section 3,4,5).

**Reproducibility:**

4: Could mostly reproduce the results, but there may be some variation because of sample variance or minor variations in their interpretation of the protocol or method.

**Reviewer Confidence:**

3: Pretty sure, but there's a chance I missed something. Although I have a good feel for this area in general, I did not carefully check the paper's details, e.g., the math, experimental design, or novelty.

**Typos Grammar Style And Presentation Improvements:**

I think there is room for improvement in the structure of the paper. For example, the data collection process (Section 6)  is presented later in the paper but should be explained before the key analysis (Section 3). Also, there is an explanation of the 21 edit types in the Appendix, but this information should also be before Section 3, otherwise, it is not easy to understand the details of the analysis.

---

> ### Author Rebuttal · Authors · 2023-08-29
>
> Thank you sincerely for your review and your dedicated time to this. We hope that our answers address your concerns and questions.
>
> ---
>
> **Responses to your comments:**
>
> > “I feel that the framework has not been fully evaluated.”
>
> The nature of a human evaluation framework in our work makes it less straightforward to be evaluated, in comparison to datasets and models that can be easily evaluated. Many existing NLP conference papers [1, 2] on human evaluation frameworks also did not and could not directly evaluate the evaluation framework itself. However, we did take the following measures to ensure our framework offers a solid and robust assessment of text simplification:
> 1. Our framework builds upon existing linguistic analyses of text simplification [3, 4] and has been refined through rounds of pilot annotations.
> 2. The annotation process follows the common practices for fine-grained evaluations [1, 2].
> 3. We include an adjudication step for span selection to resolve disagreements.
>
> [1] Yao Dou, Maxwell Forbes, Rik Koncel-Kedziorski, Noah A. Smith, and Yejin Choi. Is GPT-3 text indistinguishable from human text? scarecrow: A framework for scrutinizing machine text. ACL 2022.
>
> [2] Tanya Goyal, Junyi Jessy Li, and Greg Durrett. SNaC: Coherence error detection for narrative summarization. EMNLP 2022.
>
> [3] Sanja Štajner. New data-driven approaches to text simplification. Ph.D. thesis, University of Wolverhampton. 2016.
>
> [4] Rémi Cardon, Adrien Bibal, Rodrigo Wilkens, David Alfter, Magali Norré, Adeline Müller, Watrin Patrick, and Thomas François. Linguistic Corpus Annotation for Automatic Text Simplification Evaluation. EMNLP 2022.
>
> > “I think there needs to be a comparison of the SALSA evaluation process and the process of annotating the 21 edit types directly”
>
> We designed SALSA with a decision tree to reduce the cognitive load on a single annotation, as 21 linguistic types can be overwhelming when annotated on every edit. Our decision tree-based typology is also grounded in prior work [5] that organizes simplification into atomic operations. To make sure annotators understand and differentiate each edit type, we included explicit training for each type in our tutorial.
>
> [5] Regina Stodden and Laura Kallmeyer. TS-ANNO: An Annotation Tool to Build, Annotate and Evaluate Text Simplification Corpora. ACL 2022.
>
> > “The evaluation of the framework is also affected by the reliability of the data collected in this study.”
>
> As noted in Appendix C, edit-level selection is an inherently difficult task due to the high granularity of the framework. But our token-level agreement falls within a range similar to other fine-grained evaluation framework [6]. To further improve reliability, we perform an adjudication step (L518-L523) to resolve the disagreements in span selection.
>
> To supplement the reliability of our annotations, we calculate the inter-annotator agreement on the overall simplification score that is derived from SALSA annotation (Appendix A.3). It achieves Krippendorff’s alpha of 0.38, compared favorably to the 0.32 by rank and rate, 0.25 by direct assessment, and 0.23 by Likert scale, as reported in [7].
>
> [6] Tanya Goyal, Junyi Jessy Li, and Greg Durrett. SNaC: Coherence error detection for narrative summarization. EMNLP 2022.
>
> [7] Mounica Maddela, Yao Dou, David Heineman, Wei Xu. LENS: A Learnable Evaluation Metric for Text Simplification. ACL 2023.
>
> ---
>
> **Answers to your questions:**
>
> > “How did you adapt COMET-MQM for the simplification task?”
>
> In Table 1, We use COMET-MQM with no changes to its training or model, using the two human-written references as simplification references. We decided to include this learnable metric in our evaluation as it did surprisingly well against baselines like BLEU and SARI.
>
> *Presentation Improvements*
>
> Thanks for your suggestions! Due to the space constraints, we made a tradeoff: discussing the most interesting analysis and experiments in the main text, while moving the extensive details of our proposed framework including the motivation and definition of each edit type to the Appendix. If published, we will use the extra page to bring these details, which we are also excited about, back to the main text and move data collection (Section 6) to precede our analysis. It is an easy fix that will greatly improve the reading flow.

---

### Official Review · Reviewer_AAK9 · 2023-08-10

**Soundness:** 4

**Excitement:**

3: Ambivalent: It has merits (e.g., it reports state-of-the-art results, the idea is nice), but there are key weaknesses (e.g., it describes incremental work), and it can significantly benefit from another round of revision. However, I won't object to accepting it if my co-reviewers champion it.

**Missing References:**

Cumbicus et al. (2021) also investigated the successes and failures of ATS systems, with a very different approach.

Cumbicus-Pineda, O. M., Gonzalez-Dios, I., & Soroa, A. (2021). Linguistic Capabilities for a Checklist-based evaluation in Automatic Text Simplification. In CTTS@ SEPLN.

**Paper Topic And Main Contributions:**

This article addresses the complex challenge of evaluating automatic simplification systems. The paper is very ambitious and proposes several interesting contributions. First, it introduces a dataset which, to my knowledge, does not exist, namely a set of original and simplified sentences (produced by humans, but also by different LLMs) in which the good and bad simplifications have been annotated by 6 annotators. The annotators also judged, for each edit operation, its efficacy (for good transformations) or its severity (for bad ones). Second, the authors have integrated these assessment principles, including a typology of 21 linguistic operations, into a framework for manual evaluation of ATS, called SALSA. This framework aims at supporting more reliable human evaluation of ATS systems in the future (but this has not been tested). Finally, building on QE research on evaluation, they used the human judgments in their annotated data to train SALSA-LENS, a reference-free simplification metric that perform better, on their test set, than the other automatic metrics in the field.

To conclude, I believe this paper is very valuable for the field, as it is packed with a lot of interesting ideas and results, but it really suffers from a lack of space to expose all theses ideas in a clear way and from a poor structure. It could benefit from being rewritten and submit as a journal paper (because the authors would get more space).

**Questions For The Authors:**

- If I got it right, only spans with an edit get an efficacy or severity judgment. Then, how do you get a score (ranging from -3 to 3) for all words when training LENS-SALSA ?
- I noticed that the SALSA interface displays the whole original and simplified sentences. In such setting, you did not discuss to which extent it is possible, for a human annotator, to assess the  efficacy or severity of a given simplification without being influenced by the rest of the sentence. In my opinion, it is not so obvious. Could you please elaborate a bit about this matter.

**Reasons To Accept:**

- This paper includes several innovative ideas that might help the field of ATS to progress. For instance, combining finer annotations of simplification operations (bad or good) with human judgments about the impact of those annotations is very interesting. In addition, this combined information then allows to train an evaluation metric that could pay more attention to successes and failure of ATS systems than previous metrics.
- I believe this paper could be an influential one, if make simpler to read (with no pun intended)
- The methodology used is solid and I have not noticed any major issue.

**Reasons To Reject:**

- The paper is really really dense. The authors have a lot to say – which is good -, but, in my opinion, it does not fit to conference format. There are about 18 pages of appendices, which do not only include supplementary material, but also important elements of the paper, such as a discussion about the annotation process and inter-annotator agreement scores, parts of the description of the SALSA-LENS model that are necessary to understand its principles, etc.
- In addition, I found the paper poorly structured. It starts by the description of the framework, which is fine, but the framework involves the description and motivation of the typology used and the data annotated. However, the typology is not motivated and its description is done in the appendices. The reader has to wait for section 6 to see the data introduced. It is also not very clear why the paper reports experimentation about SALSA adapted to QE, as this idea falls outside the proposed framework. Maybe this section could be kept for a future publication.
- The typology used is not motivated in relation to previous work in the field of ATS. Why are morphological transformations (such as tense modification) included in the “Structure” category? Is the “Conceptual” category better than the traditional “Semantic” one? What about discourse aspects; are they overlooked?
- Several scholars – including Stajner (2021) or Rennes (2022) ) - have recently stressed that we need to consider the specific needs of the different audiences for which ATS could be used. In this paper, the authors seem to overlook this fact and they propose an evaluation metric that should fit all types of texts and audiences, even though their training data has been annotated by 6 undergraduate with no experience of any specific audiences.

Rennes, E. (2022). Automatic Adaptation of Swedish Text for Increased Inclusion (Doctoral dissertation, Linköping University Electronic Press).
Štajner, S. (2021). Automatic text simplification for social good: Progress and challenges. Findings of the Association for Computational Linguistics: ACL-IJCNLP 2021, 2637-2652.


**Reproducibility:**

4: Could mostly reproduce the results, but there may be some variation because of sample variance or minor variations in their interpretation of the protocol or method.

**Reviewer Confidence:**

4: Quite sure. I tried to check the important points carefully. It's unlikely, though conceivable, that I missed something that should affect my ratings.

---

> ### Author Rebuttal · Authors · 2023-08-29
>
> Thank you sincerely for your review and your dedicated time to this. We hope that our answers address your concerns and questions.
>
> ---
>
> **Responses to your comments:**
>
> > “The paper is really really dense. The authors have a lot to say – which is good -, but, in my opinion,it does not fit the conference format.”
>
> We appreciate your acknowledgement of the depth of our paper. Our paper is centered around a fine-grained evaluation framework, while offering both analysis and modeling. We aim for a high-quality conference publication that provides a comprehensive understanding and answers multiple research questions, so we choose not to slice our paper into two. We will also take advantage of the extra page for publication, as detailed below.
>
> *Writing and Structure*
>
> Due to the space constraints, we made a tradeoff: discussing the most interesting analysis and experiments in the main text, while moving the extensive details of our proposed framework including the motivation and definition of each edit type to the Appendix. If published, we will use the extra page to bring these details, which we are also excited about, back to the main text and move data collection (Section 6) to precede our analysis. It is an easy fix that will greatly improve the reading flow.
>
> > “Several scholars have recently stressed that we need to consider the specific needs of the different audiences … the authors seem to overlook this fact and they propose an evaluation metric that should fit all types of texts and audiences, even though their training data has been annotated by 6 undergraduates with no experience of any specific audiences.”
>
> We fully acknowledge the importance of audience-specific needs and have mentioned this in lines 288-290 and 1458-1463 of our paper. Our SALSA framework is designed as a flexible framework that can be adapted to various use-cases. One important thing to notice is that the complex sentences in our dataset are significantly complex, averaging 37.3 words in length. Such complexity poses challenges even for our undergraduate annotators, making our annotations and evaluation metric broadly applicable to the general public who may also find such texts difficult to understand.
>
> > “Why are morphological transformations (such as tense modification) included in the “Structure” category? Is the “Conceptual” category better than the traditional “Semantic” one? What about discourse aspects; are they overlooked?”
> We classify morphological transformations like tense modifications under the "Structure" category because they often entail changes that go beyond individual words and affect sentence syntax, e.g., multiple verbs within a sentence. It also helps annotators to discern cases where a word is substituted with a simpler synonym, but in a different tense—such as changing "anticipate" to "expected."
>
> Our terminology of “Conceptual Simplification” follows from recent work in simplification [1,2] and is interchangeable with “Semantic Simplification.”
>
> Discourse preserving or modifying edits (as defined by Siddharth, 2006 [3]) are captured in our “Structure” edit type, as these often require either syntactic modifications or multi-edit changes. To improve the identification of these structure/discourse edits, annotators further labeled them into the 5 types listed in Table 5.
>
> [1] Sanja Štajner. New data-driven approaches to text simplification. Ph.D. thesis, University of Wolverhampton. 2016.
>
> [2] Sian Gooding. On the Ethical Considerations of Text Simplification. SLPAT. 2022.
>
> [3] Advaith Siddharthan. Syntactic Simplification and Text Cohesion. Res Lang Comput. 4, 77–109. 2006.
>
> ---
>
> **Answers to your questions:**
>
> > “How do you get a score (ranging from -3 to 3) for all words when training LENS-SALSA?”
>
> The scores for words that are part of an edit correspond to the severity (negative) or efficacy (positive) rating of that particular edit, which falls within the [-3, 3] range. Words that are not part of any edit are assigned a score of 0.
>
> > “To which extent it is possible, for a human annotator, to assess the efficacy or severity of a given simplification edit without being influenced by the rest of the sentence?”
>
> The capability to assess each simplification edit individually stems from our edit selection process, where each edit is selected so that it can be evaluated independently of the rest of the sentence. To achieve this, we implement the following measures:
> 1. We allow for overlapping edits and instruct annotators to create an edit for each individual operation (e.g., a single span could contain a re-order and substitution, yet both operations are selected and evaluated independently).
> 2. Our composite edits, structure and split edits, are intentionally designed to group alike edits to be evaluated together.
> 3. Our annotator adjudication step ensures edits are selected with the correct scope (i.e., each edit represents only one simplification operation).

---

### Official Review · Reviewer_v4Lp · 2023-08-11

**Typos Grammar Style And Presentation Improvements:** 1. Would move section 6 early on to g…
**Soundness:** 3

**Excitement:**

3: Ambivalent: It has merits (e.g., it reports state-of-the-art results, the idea is nice), but there are key weaknesses (e.g., it describes incremental work), and it can significantly benefit from another round of revision. However, I won't object to accepting it if my co-reviewers champion it.

**Paper Topic And Main Contributions:**

The task addressed in this paper is the text simplification task. The authors develop an interactive interface to collect annotations of simplifications generated by humans and various models. Through this interface, they were able to collect fine-grained information about the edit operations and their quality. With this data, they trained an evaluation metric LENS-SALSA. Further they trained models to predict the word-level quality in a supervised fashion, previously missing due to lack of appropriate annotated data.


**Questions For The Authors:**

1. The authors mentioned that there are 21 quality and error edit types. It's hard to follow where this number comes from -- there are 6 types of edit operation, each with 4 types of information change, organized into lexical, syntax, and conceptual edits. L:161 - “After being categorized into lexical, syntax, or conceptual edit families, we further classify each edit operation into 21 fine-grained success (quality), failure (error)” -- does this mean the previous edit operations and information change are separate from the 21? Appendix A provides Table 4 which provides some insight but the main paper does not reference this in the running text. Clarity on where the 21 types would help.

2. The authors use only 6 annotators for their annotation. Also for the 6 annotators, what is the significance level? It would help to justify using only 6 annotators if their significance for the annotator agreement is high.

3. The overall motivation for the text simplification task: Since there is a lot of subjectivity in how edits are made, curious if the authors referred to any Cogsci literature to understand how humans abstract details. This is an existing NLP task, but the introduction does not really motivate how this might be useful


**Reasons To Accept:**

The authors thoroughly analyze the types of edits, and their categories, and further their quality/error analysis. It is interesting to analyze how humans make edits, what types of content are worth preserving, and what content is redundant. Further, this is a contrast against the numerous baselines compared.


**Reasons To Reject:**

The authors collected a large-scale annotation of fine-grained edit information for the task of text simplification. However, the use case of the text simplification task and the data is unclear and not addressed. The flow of the paper is quite confusing and the content can be restructured better. The authors did analyze the types of errors by models and humans, but did not try to hypothesize why humans were better at some edits, why models perform the way they are, etc. There was not a sufficient interpretation of the edit analysis.

**Reproducibility:**

3: Could reproduce the results with some difficulty. The settings of parameters are underspecified or subjectively determined; the training/evaluation data are not widely available.

**Reviewer Confidence:**

4: Quite sure. I tried to check the important points carefully. It's unlikely, though conceivable, that I missed something that should affect my ratings.

---

> ### Author Rebuttal · Authors · 2023-08-29
>
> Thank you sincerely for your review and your dedicated time to this. We hope that our answers address your concerns and questions.
>
> ---
>
> **Responses to your comments:**
>
> _Writing and Structure_
>
> Due to the space constraints, we decided to discuss the most interesting analysis and experiments in the main text, while moving the extensive low-level details of our proposed framework for manual quality and error categorization including the definition of each edit type to the Appendix. If published, we will use the extra page to bring these details, which we are also excited about, back to the main text and move data collection (Section 6) to precede our analysis. It is an easy fix that will greatly improve the reading flow.
>
> > “The use case of the text simplification task and the data is unclear and not addressed.”
>
> The use case of text simplification is well-documented in existing literature [1, 2, 3], aiming to make complex text more accessible for people including but limited to children, those with cognitive disabilities, and second-language learners. Given this established understanding of the task, we chose not to reiterate its use case in our paper. However, we agree on the value of establishing the context and will elaborate on the task’s use case in the final version.
>
> As for data’s use cases, we outline them on lines 70-74 in the introduction and elaborate in Sections 3,4, and 5, which are providing a comprehensive analysis of modern large language models and automatic metrics, training a new reference-free automatic metric, and introducing the word-level quality estimation task.
>
> [1] Carroll, John, Guido Minnen, Yvonne Canning, Siobhan Devlin, and John Tait. Practical simplification of English newspaper text to assist aphasic readers. AAAI 1998.
>
> [2] J. D. Belder and Marie-Francine Moens. Text simplification for children. SIGIR 2010.
>
> [3] Sanja Stajner. Automatic Text Simplification for Social Good: Progress and Challenges. Findings of ACL 2021.
>
> > “There was not a sufficient interpretation of the edit analysis” / “did not try to hypothesize why humans were better at some edits, why models perform the way they are, etc”
>
> Our aim is to provide an empirical analysis of the behaviors we observed in humans and models’ simplifications while avoiding making indiscriminate hypotheses. In Section 3, "Key Analysis," we do discuss where the data allows for interpretation, such as the elaborative tendencies of GPT-3.5 and the conservative nature of fine-tuned T5 models. We also provide additional findings in Appendix D on specific error types and simplified techniques used by humans and models. We believe this provides a balanced and thorough analysis.
>
> ---
>
> **Answers to your questions:**
>
> > “The authors mentioned that there are 21 quality and error edit types … Clarity on where the 21 types would help.”
>
> Thank you for pointing this out. The full list can be found in Table 4, as well as Figure 7, in the Appendix, which we will add more explicit references at suitable places in the final version.
>
> > “The authors use only 6 annotators for their annotation. For the 6 annotators, what is the significance level?”
>
> These annotators are in-house annotators who went through extensive training and discussion sessions, and provide higher quality annotations than the crowdsourcing workers. The number of annotators is on par with related studies [4,5].
>
> To provide more detail to the token-level Krippendorff's Alpha agreement reported in Table 3, we perform a bootstrap analysis with 1000 iterations and report the 95% CI below:
>
> | Edit             | Sub-type           | Kripp. α 95% CI  |
> |:-----------------|:-------------------|:----------------:|
> | Insertion        | More Information   | [0.392, 0.512]     |
> | Deletion         | Less Information   | [0.722, 0.777]     |
> | Substitution     | More Information   | [0.047, 0.256]     |
> |                  | Less Information   | [0.271, 0.347]     |
> | Reorder          | Word-level         | [0.075, 0.163]     |
> |                  | Component-level    | [0.363, 0.461]     |
> | Split            | Sentence Split     | [0.625, 0.696]     |
> | Structure        | Structure          | [0.214, 0.286]     |
> | Substitution     | Same Information   | [0.500, 0.560]     |
>
> We also calculate the inter-annotator agreement on the overall simplification score that is derived from SALSA annotation (Appendix A.3). It achieves Krippendorff’s alpha of 0.38, compared favorably to the 0.32 by rank and rate, 0.25 by direct assessment, and 0.23 by Likert scale, as reported in [6].
>
> [4] Tanya Goyal, Junyi Jessy Li, and Greg Durrett. SNaC: Coherence error detection for narrative summarization. EMNLP 2022.
>
> [5] Philippe Laban, Jesse Vig, Wojciech Kryscinski, Shafiq Joty, Caiming Xiong, Chien-Sheng Wu. SWiPE: A Dataset for Document-Level Simplification of Wikipedia Pages. ACL 2023.
>
> [6] Mounica Maddela, Yao Dou, David Heineman, Wei Xu. LENS: A Learnable Evaluation Metric for Text Simplification. ACL 2023.
>
> > “Curious if the authors referred to any Cogsci literature to understand how humans abstract details.”
>
> Our framework primarily builds upon existing linguistic analyses of text simplification [7, 8]. While there is a subfield called cognitive simplification [9, 10], we didn’t delve into that area in this paper. We appreciate this suggestion and agree that connecting our work to cognitive science would be an interesting direction for future research.
>
> [7] Sanja Štajner. New data-driven approaches to text simplification. Ph.D. thesis, University of Wolverhampton. 2016.
>
> [8] Rémi Cardon, Adrien Bibal, Rodrigo Wilkens, David Alfter, Magali Norré, Adeline Müller, Watrin Patrick, and Thomas François. Linguistic Corpus Annotation for Automatic Text Simplification Evaluation. EMNLP 2022.
>
> [9] Shira Yalon-Chamovitz, Ruth Shach, Ornit Avidan-Ziv, and Michal Tenne Rinde. The call for cognitive ramps. Work, 53(2):455–456. 2016.
>
> [10] Chamovitz, Eytan, and Omri Abend. Cognitive Simplification Operations Improve Text Simplification. CoNLL 2022.

---

### Meta-Review · Area_Chair_q67V · 2023-09-14

**Recommendation:** 5

**Metareview:**

The authors built an interactive interface to collect annotations of simplifications made by humans and models. Using this data, they developed an evaluation metric called LENS-SALSA and trained models to predict word-level quality. All the reviewers unanimously commend the authors for their meticulous examination of edit types, their categorization, and the extensive analysis of the data collected by SALSA.

However, the major flaw of the paper, as pointed out by the reviewers, is the excessive volume of information it presents while pushing substantial and pertinent content to the appendix due to space constraints. Reviewers express dissatisfaction with the paper's intricate structure and concur, as humorously noted by reviewer AAK9, that it requires simplification.
I am confident that the authors have the capacity to implement revisions to the paper within the provided timeframe, including the extra page. My strong recommendation, however, is for them to earnestly consider the invaluable guidance from the reviewers regarding the paper's restructuring.

---

### Decision · Program_Chairs · 2023-10-07

**Decision:**

Accept-Main

**Comment:**

The authors built an interactive interface to collect annotations of simplifications made by humans and models. Using this data, they developed an evaluation metric called LENS-SALSA and trained models to predict word-level quality. All the reviewers unanimously commend the authors for their meticulous examination of edit types, their categorization, and the extensive analysis of the data collected by SALSA.

However, the major flaw of the paper, as pointed out by the reviewers, is the excessive volume of information it presents while pushing substantial and pertinent content to the appendix due to space constraints. Reviewers express dissatisfaction with the paper's intricate structure and concur, as humorously noted by reviewer AAK9, that it requires simplification.
I am confident that the authors have the capacity to implement revisions to the paper within the provided timeframe, including the extra page. My strong recommendation, however, is for them to earnestly consider the invaluable guidance from the reviewers regarding the paper's restructuring.